# Evidence for 4e charge of Cooper quartets in a biased multi-terminal graphene-based Josephson junction

Ko-Fan Huang[1,6], Yuval Ronen [1,6], Régis Mélin[2], Denis Feinberg[2], Kenji Watanabe [3], Takashi Taniguchi [4] & Philip Kim [1,5✉]

In a Josephson junction (JJ) at zero bias, Cooper pairs are transported between two superconducting contacts via the Andreev bound states (ABSs) formed in the Josephson channel. Extending JJs to multiple superconducting contacts, the ABSs in the Josephson channel can coherently hybridize Cooper pairs among different superconducting electrodes. Biasing three-terminal JJs with antisymmetric voltages, for example, results in a direct current (DC) of Cooper quartet (CQ), which involves a four-fermion entanglement. Here, we report half a flux periodicity in the interference of CQ formed in graphene based multi-terminal (MT) JJs with a magnetic flux loop. We observe that the quartet differential conductance associated with supercurrent exhibits magneto-oscillations associated with a charge of 4e, thereby presenting evidence for interference between different CQ processes. The CQ critical current shows non-monotonic bias dependent behavior, which can be modeled by transitions between Floquet-ABSs. Our experimental observation for voltage-tunable non-equilibrium CQ-ABS in flux-loop-JJs significantly extends our understanding of MT-JJs, enabling future design of topologically unique ABS spectrum.

[1] Department of Physics, Harvard University, Cambridge, MA 02138, USA. [2] Université Grenoble—Alpes, CNRS, Grenoble INP, Institut NEEL, 38000 Grenoble, France. [3] Research Center for Functional Materials, National Institute for Materials Science, 1-1 Namiki, Tsukuba 305-0044, Japan. [4] International Center for Materials Nanoarchitectonics, National Institute for Materials Science, 1-1 Namiki, Tsukuba 305-0044, Japan. [5] John A. Paulson School of Engineering and Applied Sciences, Harvard University, Cambridge, MA 02138, USA. [6]These authors contributed equally: Ko-Fan Huang, Yuval Ronen. ✉email: pkim@physics.harvard.edu

At a normal (N)-superconductor (S) boundary, below the superconducting gap $\Delta$, current is induced via Andreev reflection (AR)[1] i.e., an electron impinging on S binds to another electron near the interface, transmitting a Cooper pair into the S region while a hole is reflected. By constructing two such boundaries one creates an SNS Josephson junction (JJ), which can be viewed as an electronic analog of the optical Fabry-Perot interferometer: Each boundary acts as an AR mirror and resonances are formed in the junction. In this case, these resonances coherently superimpose electron and hole waves, forming the so-called Andreev bound states (ABSs)[2–4]. Each AR picks up the phase of the corresponding superconductor; therefore, the ABS wave-functions and energies depend on the phase difference $\varphi$ between the two superconductors. Each populated ABS $\alpha$ contributes a current, derived from its ABS energy $E_\alpha(\varphi)$ with respect to $\varphi$, to the total Josephson current. Recently, much attention has been paid both in theoretical predictions[5–7] and in experiments[8–12] to multi-terminal Josephson junctions (MT-JJs), where a single metallic region bridges three or more superconductors. With ARs taking place at each SN interface, the SNS physics is generalized in several aspects. First, the equilibrium ABS spectrum of a multi-terminal JJ depends on multiple phase differences $\varphi_i$, where $\varphi_i$ is the phase of $i$th S electrodes connected to the junction[13,14]. The ABS energy appears as a contour in a multi-dimensional voltage space[11,12]. The high dimension phase space spanned by $\varphi_i$'s offers the prospect of engineering artificial high-dimensional crystal band structures with topological properties[15–19]. Second, multi-dimensional current-voltage characteristics may present a complex subgap structure due to local (between two terminals) or non-local (among multiple terminals) multiple Andreev reflections (MARs)[9–12,20]. Third, MT-JJs allow DC transport of multiple entangled Cooper pairs for commensurate combinations of applied voltages. For instance, in a three-terminal junction with two leads biased with anti-symmetric bias scheme at $(V, -V)$, a Cooper pair from the grounded terminal is split into two quasiparticles via crossed AR[21–23]. The two quasiparticles propagate toward two distinct terminals and then recombine with the ones originated from the other pair splitting, forming two entangled Cooper pairs—the Cooper quartet (CQ) within the junction[6,7] (see Fig. 1a). We note that the crossed AR is a local AR at one of the SC electrodes as opposed to CAR across a SC metal.

Since a CQ minimally requires four coherent ARs, its underlying mechanism is distinct from a simple extrinsic locking between two separated JJs biased at opposite voltages. In the externally coupled JJs with the antisymmetric bias condition, this can produce an alternating current (AC) Josephson oscillations with the same frequency. Synchronization of these oscillations can occur by photon exchange between the JJs via a classical impedance[24]. This view of mode-locked JJs, however, only considers external coupling between local AR processes. For MT-JJs with low energy ABS in the weak link, new possibility arises for an intrinsic synchronization of asymmetrically biased JJs via non-local AR processes[6,7], leading to the entangled CQ spreading over multiple JJs.

MT-JJs with conducting weak links have been fabricated in 2D metallic[8] and 1D semiconducting[9] channels as well as in graphene[10]. While non-local supercurrent was probed in MT-JJs by measuring cross-correlated current noise[9], a direct experimental observation of the presence of phase coherent entangled CQs has yet to be realized. In this work, we employ a magnetic flux loop coupled to the MT-JJs to modulate the junction properties. Using both bias voltage and threaded magnetic flux, we control the CQ dynamics, including coherent CQ-ABS and interference between different CQ processes. As the bias increases, we find non-monotonic behavior of the CQ critical current as

a function of bias, which can be interpreted within a simplified model by transitions between Floquet CQ-ABSs generated by intrinsic synchronizing of the entangled CQs[25–27].

## Results and discussions

**Characterization of MT-JJ in the Josephson regime.** Along with the ability of controlling the number of conducting channels, low superconducting contact resistance and weak back-scattering[28–30] make graphene an ideal choice for exploring MT-ABS physics. Utilizing the tunability of graphene chemical potential, one can modulate the coupling strength at each contact, thereby engineering the ABS spectrum. Our graphene-based MT-JJs use Ti/Al as the superconducting contacts, where Al is chosen owing to its large superconducting coherence length (~1 μm). A three-terminal JJ (four contacts including a superconducting loop) is fabricated on the graphene-hBN-SiO$_2$ structure as shown in Fig. 1a (additional fabrication information can be found in the "Method" section).

All measurements were performed at 300 mK. Before we conduct the MT-JJ measurement, we first characterized our device with a two-terminal measurement as the S-loop implements a superconducting quantum interference device (SQUID) geometry. For this measurement, we applied a bias voltage $V$ to the loop via two series connected RC filters. The output current $I$ is measured at $S_2$ while $S_1$ is floating. Figure 1c shows an $I$-$V$ measurement curve of the junction. In the smaller bias regime, supercurrent flows in the junction and the bias voltage drops are only on the series connected resistors $R_{RC}$ (200 Ω each) in the filters. As the current exceeds the critical current $I_c$ of the SQUID, the slope of $I$-$V$ curve changes suddenly at the corresponding applied voltage $V_c$. Since the bias voltage is distributed among two filter resistors and the normal junction resistance, the critical current can be obtained from $I_c = V_c/2R_{RC}$. Upon applying the magnetic field $B$, $I_c$ is modulated and exhibits SQUID-like pattern as a result of the two interfering superconducting paths in the loop (blue and red dashed lines in Fig. 1b). Figure 1d shows the differential conductance ($G = dI/dV$) as a function of bias voltage and magnetic field. The higher conductance area near the zero-bias regime (central part) is the supercurrent region and its edges mark the value of $I_c$. As the magnetic field is swept, $I_c$ is modulated with a periodicity of $\delta B = 145$ μT, corresponding to the unit flux quantum $\Phi_0 = h/2e$ for an enclosed area of $A = 14.2$ μm$^2$, matching our device loop size (including the area increase due to London penetration depth). An additional lower frequency ($\delta B_F = 3$ mT) originated from the Fraunhofer oscillations is observed, corresponding to an area of 0.69 μm$^2$, which agrees with the junction dimensions. We find that the strength of the critical current can also be tuned according to the graphene carrier density via a back-gate voltage $V_{bg}$. As shown in Fig. 1e, $I_c$ decreases monotonically as $V_{bg}$ approaches the charge neutrality point of graphene located at $V_{bg} \approx -32$ V. Reduction of $I_c$ close to the Dirac point is expected due to the decreasing number of ABS carrying current in the graphene channels[31].

**Cooper quartet.** With reconfiguration of the external circuitry, our device can serve as a MT-JJ where the common N-region graphene channel is proximitized. MT-JJ with magnetic flux loops was studied theoretically and experimentally in bi-SQUID devices[32–34], where the equilibrium (i.e., no potential difference between the junctions) ABS spectrum was investigated. Our four-contact device geometry with gate-tunable graphene weak link allows us to study biased MT-JJs in the non-equilibrium regime, where the non-local CQ formation can be investigated[35]. Moreover, by threading a flux through the device loop we aim to modulate the CQ-ABS spectrum. Figure 2a shows the

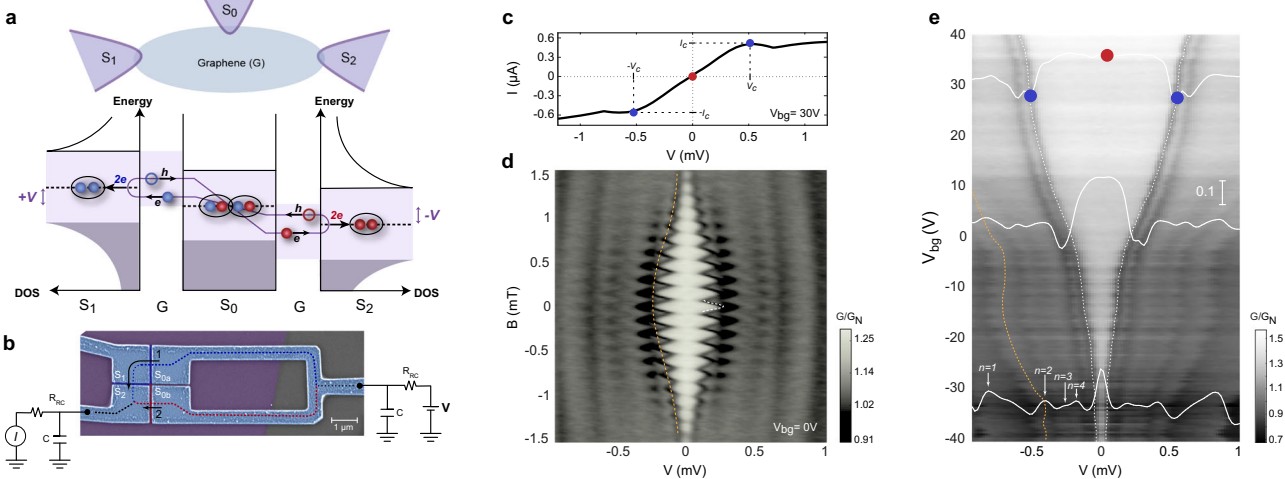

**Fig. 1 Illustration of quartet formation and a single source voltage bias characterization of four-terminal Josephson junction including a loop.**
**a** Schematic illustration of the three-terminal quartet process with the Andreev reflection picture. The middle superconductor $S_0$ is grounded while the other two superconductors are biased at $+V$, $-V$, respectively. The two entangled Cooper pairs (with red and blue electrons) are formed in $S_0$ through two local Andreev reflections and two crossed Andreev reflections. **b** False color scanning electron microscopy (SEM) image of the device with measurement configurations. Graphene (purple) is top-contacted by Ti/Al superconducting electrodes (blue) and the electrode separations typically are 80–100 nm. Here we split $S_0$ in **a** into two contacts $S_{0a}$, $S_{0b}$ connected by a loop. **c** $I - V$ curve of the device from the measurement configuration in **b** $I_c$ is the critical current and the corresponding voltage value is labeled as $V_c$ (the blue dots). **d** Magnetic field dependence, $dI/dV$ as a function of the bias voltage and magnetic field. Bright region (high conductance) is the supercurrent and the edge corresponds to the value of critical current, which is modulated by the magnetic field. The SQUID-like pattern indicates the interference between two supercurrent paths (red and blue dashed lines in **b**). The periodicity of the fast oscillation (white dashed curve) corresponds to the loop area and the slow oscillation (yellow dashed curve) is the first lobe of Fraunhofer pattern. **e** Gate dependence of the supercurrent, $dI/dV$ as a function of the bias voltage and global back-gate voltage $V_{bg}$. The critical current reaches the minimum as graphene is tuned to the Dirac point near $V_{bg} = -32$ V.

measurement scheme adopted in this study for phase sensitive quartet detection. We apply DC bias voltages $V_1$ and $V_2$ to $S_1$ and $S_2$, respectively, and a small AC bias voltage $\delta V$ to the loop electrodes $S_{0a}$ and $S_{0b}$. At given bias voltages, we measure the AC current contributions $\delta I_1$ and $\delta I_2$ flowing to $S_1$ and $S_2$, respectively. Voltages $V_1$ and $V_2$ are applied to the total circuit including $R_{RC}$, which is about 100 times larger than the actual voltage applied at the junction (see circuit in Fig. S1).

Figure 2b shows the differential conductance measured at $S_1$ ($G_1 = \delta I_1/\delta V$) as a function of the two DC bias voltages $V_1$ and $V_2$ with $V_{bg} = 40$ V. We identify four high conductance regions (marked by four white dashed lines crossing at the origin), which correspond to four different supercurrents. For instance, when $S_2$ and $S_0$ are equipotential along $V_2 = 0$, a Josephson supercurrent flows between these two contacts carried by a Cooper pair-ABS. Subfigures (i), (ii) and (iii) illustrate these local supercurrents between different pairs of S-contacts. The critical values of the supercurrents can be extracted from the widths of the signals, which are 0.47, 0.42, 0.38 μA, respectively. Similar data can be obtained for differential conductance $G_2 = \delta I_2/\delta V$ measured at $S_2$ (see Section 2 in the Supplementary Information).

In addition to the two-terminal Josephson currents (i)–(iii), we observe another supercurrent signal along the $V_1 = -V_2$ line, as shown in Fig. 2b. This line originates from the sharp black lines, which define the 2-terminal critical current contour (CCC). To ascertain the intrinsic nature of this signal and that it is due to quartets, let us first remark that no clear MARs are observed in this sample in the bias voltage range where we observe a quartet signal. Indeed, given the low value of the junction voltage, those MARs, whether local or non-local, would have very high orders. In a non-ballistic graphene with interface scattering, such high-order MARs are unlikely to take place, in contrast to clean InAs 2DEG samples such as those in ref. [11]. In the work of ref. [11], where the critical currents are high, the situation is very different:

several bright local MAR lines were observed, but no supercurrent was observed along the $V_1 = -V_2$ line beyond the CCC. This is not surprising because quartets require four ARs, two local and two non-local processes, and are easily masked by bright local MARs. Notice that the same conditions (low voltage compared to the gap, no MARs or very weak ones) were met in refs. [8,9] and a quartet line was indeed observed.

We labeled the $V_1 = -V_2$ line as (iv) Quartet and it signals the existence of non-equilibrium CQ-ABSs within the junction, despite the fact that all contacts are at different chemical potentials. In this regime all $2e$ Josephson currents taking place between each pair of terminals are AC. On the contrary, in this configuration, as depicted in Fig. 1a, two Cooper pairs from two S-contacts ($S_1$ and $S_2$) are entangled into a four-electron state via two local ARs and two crossed ARs at the middle S-contact ($S_0$) to form CQ-ABS[5–9]. The shape and width of the anomaly is very similar to that of an ordinary Josephson current, and it allows to define a "quartet critical current" $I_{qc}$. Remarkably, in this regime where the local DC-Josephson currents disappear, the CQ-ABSs form only when the junction is biased antisymmetrically and they carry non-local DC supercurrent flowing among all terminals simultaneously. The corresponding bias condition $V_1 + V_2 = 0$ satisfies the energy conservation for the CQ DC current, where correlated Cooper pairs originating from $S_1$ and $S_2$ are simultaneously transmitted into $S_0$. Notice that this necessary condition does not tell anything about the microscopic mechanism for quartets. Our experiment precisely helps elucidating this mechanism, by using the tool of a magnetic flux and by investigating the periodicity and the voltage dependence of the field modulation.

**Two types of Cooper quartet processes.** Similar quartet supercurrent signatures were inferred previously in three-terminal JJ made from diffusive metal[8] and 1D nanowires[9]. The novelty in

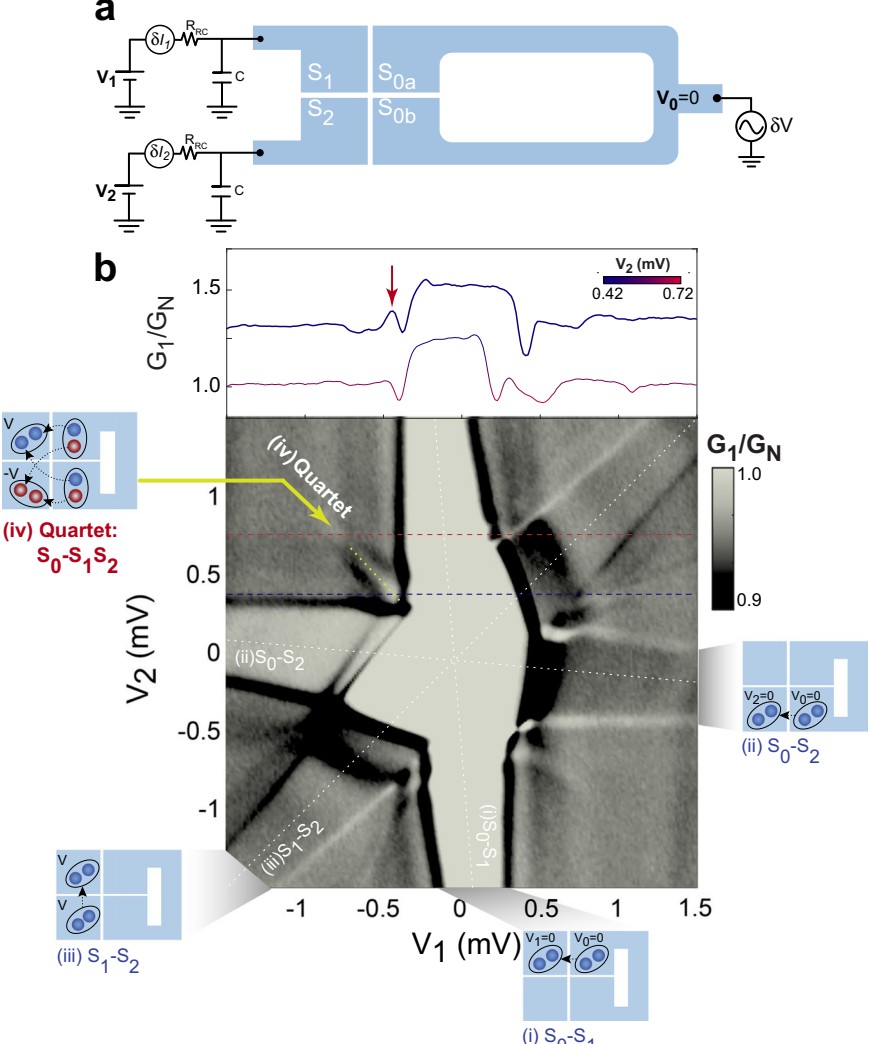

**Fig. 2 Dual source voltage bias characterization for quartet detection. a** Configuration for the quartet measurement. The loop $S_0$ is grounded while potential of the first ($S_1$) and second ($S_2$) electrodes are controlled via DC voltages $V_1$ and $V_2$, respectively. An additional AC excitation $\delta V$ is applied to the loop, and the AC current $\delta l_1$ ($\delta l_2$) through $S_1$ ($S_2$) is measured. **b** top panel: differential conductance $G_1$ ($= \delta l_1/\delta V$) measured at $S_1$ as a function of the DC bias voltage $V_1$ when $V_2$ is tuned from 0.42 to 0.72 mV. Bottom panel: color plot of $G_1$ as a function of $V_1$ and $V_2$ (excluding the circuit resistance $R_{RC}$), with $V_{bg} = 40$V. In total, there are four different supercurrents in the device. Inset (i)–(iii) show the local Josephson supercurrent between any pair of leads at the same potential. Inset (iv) Quartet shows the non-local quartet supercurrent flowing among all three superconducting leads and the quartet signal in ($V_1$, $V_2$)-plane is the narrow yellow region along the −45 degree direction (where the red arrow points at).

our four-terminal JJ device is the study of the nontrivial bias voltage dependence, and the presence of a magnetic flux loop, enabling direct probing of the CQ-ABS coherence via the periodic dependence of the critical current with magnetic field. The left panels of Fig. 3a, b show the quartet differential conductance measured at $S_2$ (i.e., $G_2$ along the quartet line $V_1 = -V_2$) as a function of the magnetic flux $\Phi = B \cdot A$ measured at different back-gate voltage $V_{bg}$. The quartet differential conductance $G_{i=1,2}$ probes the quartet critical current $I_{qc}$ (see Section 5 in the Supplementary Information). As a function of $\Phi$, clear oscillations of $G_i$ are observed, demonstrating periodic modulation of $I_{qc}(\Phi)$ due to phase coherence of the CQ-ABS. By taking the Fourier transform of $G(\Phi)$ (the right panel of Fig. 3a, b), we find two major periodicities $\Phi_0/2$ and $\Phi_0$, where $\Phi_0 = h/2e$. The relative strength of the periodicities is tuned non-monotonically, since $V_{bg}$ modifies the number of channels in graphene as well as the coupling of S-electrodes, which modifies the ABS spectrum.

In particular, at $V_{bg} = 25$ V (Fig. 3a), $I_{qc}$ exhibits a prominent contribution from $\Phi_0/2$-periodicity, which, as we show below, provides direct evidence for the charge $4e$ associated with the CQ-ABS.

At first sight, the observation of the two periodicities tuned by the gate voltage resembles the SQUID oscillation in Fig. 1d, where the $\Phi_0/2$ oscillation would be viewed as the second harmonic of the fundamental quantum flux periodicity. However, the magneto-oscillation here in Fig. 3a, b cannot be related to usual DC-SQUID harmonics since, as stressed above, we do not operate in the Josephson regime but well beyond, i.e., in a range where AC Cooper pair Josephson currents flow between each pair of terminals, rather than DC ones. Only the junction between $S_{0a}$ and $S_{0b}$ is equipotential but the current is not measured through this junction. Furthermore, as opposed to the 2-terminal case, in a MT-JJ, quasi-particle current and quartet DC current flow simultaneously due to the inequivalent chemical potential of the

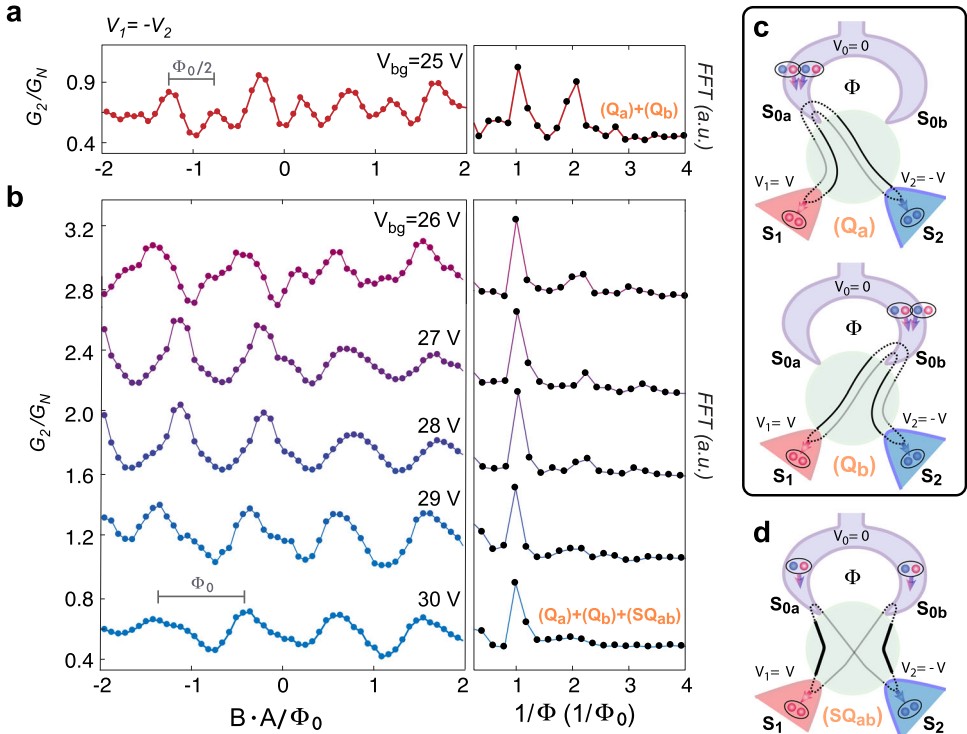

**Fig. 3 Different types of quartet process. a** Left panel shows the quartet differential conductance $G_2$ ($= \delta I_2/\delta V$) measured at $S_2$ as a function of the flux $\Phi = B \cdot A/\Phi_0$, where $V_{bg} = 25$ V. In the right panel, discrete Fourier transform (DFT) analysis of the data shows prominent harmonics, a consequence of 8 oscillations in the left panel. The quartet is biased at $V_1 = -V_2 = 0.4$ mV, where the DC 2e process and MARs are not effective. **b** for $V_{bg} = 26$–30 V. The periodicity evolves from half-flux quantum to one flux quantum as $V_{bg}$ increases. **c** ($Q_a$) ($Q_b$) shows the conventional three-terminal quartet process with only one out of the two loop contacts involved. Electron-hole conversion happens twice at the same contact of the loop (either $S_{0a}$ or $S_{0b}$), resulting in periodicity of half-flux quantum. **d** ($SQ_{ab}$) shows the split-quartet process involving both contacts of the loop. With the odd parity of Cooper pairs transferred, the periodicity is one flux quantum.

SC contacts under the quartet condition. We delineate our signal from the quasi-particle current contribution by measuring an oscillatory differential conductance, following the $I_{qc}(\Phi)$ variation along the quartet bias condition ($V_1 = -V_2$) on top of the quasi-particle current background (see Fig. S3c).

The observed periodicities are therefore intrinsic to the quartet process itself. In a perturbative model approach expanded toward the finite bias regime (see Section 5 in the Supplementary Information for detailed discussion), we consider the minimal number of four ARs required for quartet processes, taking place between four terminals instead of three. We find that the modulation of the periodicity is indeed associated with interference of three different contributions to the CQ-ABS: two conventional quartets (3-terminal), denoted as $Q_a$ and $Q_b$, and a novel process, the split-quartet (specific to four terminals), denoted as $SQ_{ab}$. As shown in Fig. 3c, the two conventional quartets, $Q_a$ and $Q_b$, take place among $S_1$, $S_2$ and only one electrode of the S-loop. In these processes, the entangled Cooper pairs enter the loop either through $S_{0a}$ or $S_{0b}$. Since every AR picks up the phase of the superconducting contact, these conventional quartet processes acquire phase factors $e^{i(\varphi_1+\varphi_2)}$ at $S_{0a}$ and $e^{i(\varphi_1+\varphi_2+4\pi\Phi/\Phi_0)}$ at $S_{0b}$, where $\varphi_1$ ($\varphi_2$) is the phase difference between $S_1$ ($S_2$) and $S_0$. Note that the factor 4 in the exponent reflects that two Cooper pairs depart from the same electrode of the grounded loop. If there were only this type of 3-terminal quartet process in the system, the phase factor at $\Phi/\Phi_0 = 0$ would become equivalent to that at $\Phi/\Phi_0 = 1/2$, leading to $\Phi_0/2$-periodicity in $I_{qc}(\Phi)$.

While the conventional quartet process $Q_a$ and $Q_b$ described above is common with simple three-terminal JJs, the three-

terminal JJ with a loop enables a new type of quartet, the split-quartet process $SQ_{ab}$. As shown in Fig. 3d, two entangled Cooper pairs are spatially separated into the two electrodes of the loop, yielding a phase factor $e^{i(\varphi_1+\varphi_2+2\pi\Phi/\Phi_0)}$. Interference of split and conventional quartet processes leads to $\Phi_0$-periodicity. We observe that the Fourier component associated with $\Phi$ periodicity stays constant while the $\Phi/2$ component varies sensitively with the gate voltage. Although a full understanding of this dependence is beyond the scope of this work, it indicates that the strengths of the two different (i.e., conventional and split) quartet processes are determined by the relative contact couplings, which are tunable via gating (see Section 5 in the Supplementary Information).

**Bias voltage dependence of quartet supercurrent.** Most importantly, the quartet supercurrent can be modulated by the quartet voltage $V_q$, which is the actual voltage applied on the junction along the $V_1 = -V_2$ line. The variation of $G(\Phi, V_q)$ with $\Phi$ or $V_q$ is proportional to that of $I_{qc}(\Phi, V_q)$ along the quartet line since it is an increasing function of the critical current (see Section 5 in the Supplementary Information). Therefore, this differential conductance measurement serves as a good indicator to investigate the behavior of quartets as a function of magnetic field and the quartet voltage $V_q$. Figure 4a shows a 2D color plot of $G_1$ (the quartet conductance measured at $S_1$) as a function of $V_q$ and the normalized magnetic flux $\Phi/\Phi_0$ at a fixed gate voltage $V_{bg} = -5$ V, where the quartet current is strong (see Fig. Supplementary 4 in the Supplementary Information). Both voltage scales (at the junction and at the circuit resistance) are presented,

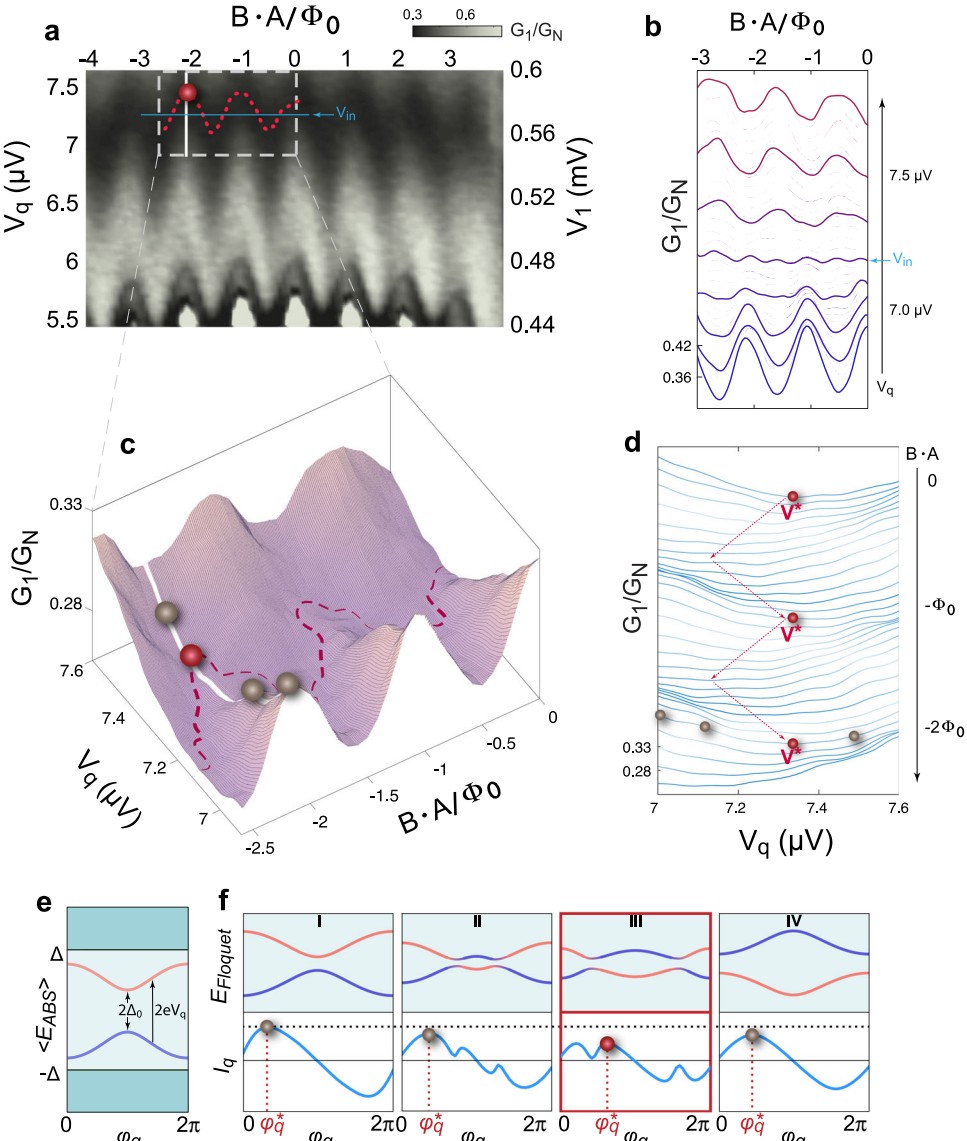

**Fig. 4 Quartet conductance and the Floquet spectrum as a function of $V_q$ and magnetic flux. a** Quartet conductance $G_1$ measured along the quartet line in $(V_q, \Phi)$-plane. Left y-axis is quartet voltage, taken across the junction. Right y-axis is the external applied voltage as shown in Fig. 2 along the yellow dotted line. The red dashed line traces the minimum conductance for $-2.5 < -\Phi/\Phi_0 < 0$ and the red sphere represents the local minimum at $\Phi/\Phi_0 = -2$. **b** waterfall plot of $G_1(\Phi)$ for $V_q = 6.9$–7.6 μV. It shows clear evolution of $G_1$ from maxima to minima at integer values of flux. At the critical quartet $V_q \approx V_{in} \equiv 7.2$ μV, periodicity is $\Phi_0/2$ and for $V_q > V_{in}$, the quartet critical current is «inverted». **c** Zoom-in surface plot of $G_1(V_q, \Phi)$ for $-2.5 < \Phi/\Phi_0 < 0$. The winding of the red sphere (local minimum) is marked with the red dashed line, matching that in **a**. The gray spheres represent quartet conductance at different values of $V_q$. **d** waterfall plot of $G_1(V_q)$ for $\Phi/\Phi_0 = -2 \sim 0$. The local minimum $V^*$ presents a zig-zag pattern as flux is tuned. **e** When the quartet voltage $V_q$ is in the adiabatic limit, the adiabatic Andreev levels $<E_{ABS}>$ depend only on one phase variable, the quartet phase $\varphi_q$. The minimum difference between the two levels is the Andreev gap $\Delta_0$ and a finite $V_q$ creates resonant coupling between the two levels. **f** upper panel shows the energy of the Floquet states as a function of the quartet phase $\varphi_q$ at different values of $V_q$. The corresponding quartet current $I_q$ carried by these Floquet states is shown in the lower panel. The gray and red spheres mark the critical values of the quartet current $I_{qc} = I_q(\varphi_q^*)$, matching the ones in **c**. In (III), the red sphere denotes $V_q = V^*$ when $I_{qc}$ reaches a local minimum, reflecting an avoided crossing in the Floquet spectrum.

with the latter equal to the quartet line in the zero-field map of Fig. 2b. This shows that this voltage region lies beyond the Josephson regime (black line in the bottom of Fig. 4a). At a constant $V_q$, $G_1(\Phi)$ exhibits oscillations corresponding to $I_{qc}(\Phi)$ with periodicity $\Phi_0/2$ and $\Phi_0$ components as discussed in Fig. 3. Interestingly, we find that the oscillation period and phase of $I_{qc}(\Phi)$ are also tunable as $V_q$ varies. As shown in Fig. 4b, in the low bias regime ($V_q < 7.2$ μV), $I_{qc}(\Phi)$ shows predominantly the $\Phi_0$-periodic oscillation, in phase with the SQUID phase of equilibrium supercurrent. However, as $V_q$ increases, $I_{qc}(\Phi)$ oscillation

becomes predominantly $\Phi_0/2$-periodic near $V_q \approx V_{in} \equiv 7.2$ μV. Above this critical bias voltage $V_{in}$, $I_{qc}(\Phi)$ oscillation resumes the $\Phi_0$-period, but the phase is shifted by $\pi$ compared to $I_{qc}(\Phi, V_q < V_{in})$. For this high bias quartet regime ($V_q > V_{in}$), the flux dependence of the quartet critical current is «inverted», i.e., $I_{qc}(\Phi = 0) < I_{qc}(\Phi = \Phi_0/2)$, suggesting that an unusual quartet behavior occurs as we approach the high bias limit. Indeed, a naive expectation is that destructive interference would instead decrease the quartet critical current for half-flux in the loop. The quartet bias condition ($V_1 = -V_2$) is essential for observing this 0-π phase

change as such phase change is absent at incommensurate bias condition ($V_1 \neq -V_2$) (see Fig. Supplementary 7 in the Supplementary Information).

Furthermore, for a fixed $\Phi$, $G_1(V_q)$ displays distinct non-monotonic behavior as $V_q$ varies near $V_{in}$. As shown in Fig. 4c (zoom-in 3D map) and Fig. 4d (line-cuts for flux in the range $[-2.5\Phi_0, 0]$), $G_1(V_q)$ first decreases to yield a local minimum at $V_q \approx V^*$ (represented by the red sphere) and then increases for $V_q > V^*$. We note that $V^*$ shifts in a zig-zag pattern in the ($V_q$, $\Phi$)-plane centered at $V_{in}$. Particularly, $V^*(\Phi)$ is the largest at integer $\Phi_0$ and the smallest at half-integer $\Phi_0$, similar to the inverted quartet current in previous discussion. The non-monotonic variation of $I_{qc}\left(V_q\right)$ and the inversion of the quartet current flux dependence provide a clue to the dynamic behavior of quartets in the non-equilibrium condition at a finite $V_q$. We note that, in order to reproduce such a behavior with a phenomenological extrinsic locking MT-JJ model[24], one would need to introduce an ad hoc anomaly of the circuit impedance $Z(\omega)$ at the Josephson frequency corresponding to $V$, and moreover assume that $Z(\omega)$ is modulated by the magnetic flux, which is unlikely. Another possible cause of a voltage-dependent anomaly are Fiske steps[36], realized in a tunnel junction that is coupled to a cavity resonance. To confine these modes, a sufficiently large magnetic field is required, corresponding to the area of the JJ. In our study, the anomaly appears in a much lower field range, corresponding to the larger SC loop area. Therefore, our observation cannot be associated with the Fiske steps. Finally, loop impedance effects may also be neglected at the measured voltage ranges (~μV, ~GHz) as it is estimated at 2.5 mΩ.

**Emergence of Floquet energy bands**. To better understand the $V_q$ dependence of quartet current, we suggest considering the superconducting phase modulation due to the AC Josephson effect at finite bias. In the presence of a voltage bias $V$, the Josephson relation $\dot{\varphi}(t) = \frac{2eV}{\hbar}$ implies a periodic sweeping in time of the ABS energies, defined at equilibrium as functions of $\varphi_1$ and $\varphi_2$. We set the phase of the grounded loop $S_0$ to be zero and the other two superconducting leads $S_1$ and $S_2$ have phases $\varphi_1$ and $\varphi_2$, respectively. When voltages are applied to $S_1$ and $S_2$, the phases acquire time ($t$)-dependence following the Josephson relation: $\dot{\varphi}_1(t) = \frac{2eV_1}{\hbar}$, $\dot{\varphi}_2(t) = \frac{2eV_2}{\hbar}$. Under the quartet condition ($V_1 = -V_2$) and by choosing a new set of phase variables $\varphi_q \equiv \varphi_1(t) + \varphi_2(t)$, $\varphi_r \equiv \varphi_1(t) - \varphi_2(t) = \frac{4eV_q}{\hbar}t$, we obtain a stationary quartet phase $\varphi_q$ and a running phase $\frac{\varphi_r}{2}$ that is periodically driving the system with a frequency $\frac{2eV_q}{\hbar}$, enabling intrinsic synchronization of CQs. In the adiabatic limit, i.e., $V_q$ being much smaller than the Andreev minigaps $\Delta_0$ between the ABS pairs, one can take a time average of the equilibrium ABS spectrum over $\varphi_r$ and obtain an adiabatic ABS energy $<E_{ABS}>$, which now only depends on the quartet phase $\varphi_q$ (see Section 5 in the Supplementary Information). For simplicity, let us illustrate the effect of a running phase on the ABS spectrum by considering only a single pair of ABSs at a bias small enough for the adiabatic approximation to work. $I_q$, the supercurrent carried by quartets, can then be derived from the usual JJ current-phase relation: $I_q = \frac{2e}{\hbar}\partial <E_{ABS}>/\partial \varphi_q$. However, as $V_q$ increases, we eventually enter the non-adiabatic regime: the running phase $\frac{\varphi_r}{2}$ creates an internal effective RF-field, which triggers non-adiabatic transitions between adiabatic CQ-ABS (Fig. 4e and Supplementary Information), and thus favors the occupation of higher level ABSs. This eventually creates resonances, in a way reminiscent of

Shapiro steps[5] or microwave resonances in transparent metallic contacts[37–39], and this manifests as a quartet current minimum. This demonstrates, by analogy to other microwave resonance phenomena in metallic junctions, that a non-monotonic dependence of the quartet current on applied voltages is expected. In a set of non-equilibrium ABSs, its effective separation depends on the bias voltage in analogy to the Floquet bands[40,41] separated by $2eV_q$, emerging from the periodically driven Bloch bands[35,42].

This picture is corroborated by detailed non-equilibrium calculations within a single-level quantum dot model (see Supplementary Information Section 5Ba). While this simple model is not intended to be quantitative in a multi-channel junction as the one in our experiment, similar physics can be applied: when the Josephson frequency due to $V_q$ matches the spacing between the ABS levels, resonance would occur, resulting in an oscillation of $I_{qc}$ with $V_q$. Numerical studies on multi-channel models have confirmed this picture (see Supplementary Information Section 5Be).

Employing the Floquet energy levels $E_{Floquet}$ that are derived from a pair of $<E_{ABS}>$ biased by the quartet bias $V_q$ (Fig. 4e), we can now explain the experimentally observed non-monotonic behavior of $I_{qc}(V_q)$. Figure 4f illustrates in the simplest case of a single-channel junction the evolution of two first-order $E_{Floquet}$ as a function of quartet phase $\varphi_q$. The corresponding quartet current $I_q(\varphi_q)$, shown in the bottom panels, is obtained with the Floquet-Landau-Zener[43] consideration (see Section 5 in Supplementary Information). The critical quartet current $I_{qc}\left(= \max\left\{I_q\left(\varphi_q\right)\right\}\right)$ takes place at $\varphi_q^*$. As $V_q$ increases, four different regimes appear: (I) for $2eV_q < \Delta_0$, no resonant coupling exists between the two $<E_{ABS}>$ and the quartet current is the same as near equilibrium. (II) $2eV_q \sim \Delta_0$, i.e., the Landau-Zener (LZ)-like transitions between the two $<E_{ABS}>$ bands become appreciable, opening gaps between different Floquet bands. Hybridization between two levels and mixing of states that carry opposite directions of currents reduce the net quartet current, resulting in a drop in $I_{qc} = \max\left\{I_q\left(\varphi_q\right)\right\}$ and the shifting of $\varphi_q^*$. (III) At even larger quartet voltage $V_q = V^*$, the resonances occur at the $\varphi_q^*$ in regime (I), denting the peak in $I_q$ and thereby $I_{qc}$ reaches a minimum value. (IV) When $2eV_q$ is increased to be greater than the largest gap between the two levels, there is no more hybridization. Both the energy levels and the quartet current resume the nearly adiabatic situation, similar to regime (I). For a more accurate consideration, the non-equilibrium Keldysh formalism is also applied to multi-level ABSs[25–27]. It reveals that the inversion of $I_{qc}(\Phi)$ can be associated with the avoided crossings due to LZ transition in the Floquet bands (see Section 5 of SI). As shown in the Supplementary Information, a minimal model considers two quantum dots, each coupled principally to one of the loop electrodes. The two quantum dots are connected by matrix elements mimicking the underlying graphene layer. This model shows «inversion» in a very low $V_q$ range and in a wide range of junction parameters, consistent with the experimental observations.

**Summary**. In conclusion, we experimentally demonstrate the existence of CQ using MT-JJ with gate-tunable graphene channel. With a magnetic flux threaded through the loop in the unbiased branch of our three-terminal junction, the CQ critical oscillation exhibits two distinct $\Phi_0$ and $\Phi_0/2$ periodicities, indicating interferences between different CQ-ABS processes. At a large bias, we observe non-monotonic variation of the quartet critical

current, which can be associated to Landau-Zener tunneling between Floquet CQ-ABS levels, driven by the intrinsic effective RF-field due to the running phase of CQ.

During the publication period of our work a complementary transport signature of quartet physics on 2D InAs quantum well heterostructure have been demonstrated[44], showing the universality of the quartet ABS physics.

## Methods

**Fabrication**. The van der Waals heterostructure—monolayer graphene on top of 40–60 nm thick hBN—is assembled via the inverted stacking technique, where hBN serves as the dielectric substrate to minimize disorder[45]. The flakes are picked up through procedures similar to the dry transfer technique[46] except the order is reversed, where the bottom hBN is picked up first. Via this method the top surface of graphene is guaranteed to be clean without any polymer contact in the assembling process. The graphene layer we use in our device is larger in size compared to the MT-JJ, and have not been etched in any step of the fabrication; thereby eliminating natural/etched edge effects from interfering with the JJ transport characteristics. The superconducting contacts are made of 80 nm thick aluminum with 5 nm thick sticking layer of titanium, directly deposited on graphene through electron-beam evaporation at a pressure of low $10^{-7}$ torr. Each channel is designed to be 80–90 nm to ensure the existence of supercurrents among all of the superconductors.

**Measurement setup**. The measurements are performed in He-3 fridge with the base temperature 300 mK, well below the superconducting critical temperature of aluminum ($T_c \sim 1.1$ K) and the dual voltage source measurement scheme allows the detection of quartet signal (see Supplementary Information for additional details).

## Data availability

The data generated in this study have been deposited in the online depository Zenodo (https://doi.org/10.5281/zenodo.6549095).

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

## Acknowledgements

We thank B. Douçot for his collaboration on the Floquet theory. K.-F.H. acknowledges support from DOE (DE-SC0019300) for sample preparation and fabrication. Y.R. acknowledges support from NSF (QII-TAQS MPS 1936263) for device characterization. P.K.

acknowledges NSF (DMR1809188) for data analysis. R.M. acknowledges the use of the resources of the Mésocentre de Calcul Intensif de l'Université Grenoble-Alpes (CIMENT) and the French National Research Agency (ANR) in the framework of the Graphmon project (Grant No. ANR-19-CE47-0007). K.W. and T.T. acknowledge support from the Elemental Strategy Initiative conducted by the MEXT, Japan, Grant Number JPMXP0112101001, JSPS KAKENHI Grant Number JP20H00354 and the CREST(JPMJCR15F3), JST.

## Author contributions

Y.R., K.-F.H. and P.K. designed the experiment. P.K. supervised the project. K.-F.H. and Y.R. fabricated the devices. T.T. and K.W. provided single crystals of hBN. K.-F.H. and Y.R. performed the measurements. K.-F.H., Y.R., and P.K. analyzed the data. R.M. and D.F. carried out the theoretical modeling. K.-F.H., Y.R., and P.K. prepared the manuscript and Supplementary Information with input from all the authors.

## Competing interests

The authors declare no competing interests.
