## [Peer Review File · Nature Communications]

Reviewers' Comments:

Reviewer #1:

Remarks to the Author:

The manuscript by Huang, Ronen et al presents a timely and intriguing work on multi-terminal Josephson junctions (MT-JJs), a system that is receiving increasing attention due to the potential for realizing topological ABS spectra. The focus of this work is not on topology, but on Cooper pair entanglement (quartets) that can arise in MT-JJs. The work is very clear and although MT-JJs have been studied in graphene and other systems, there work is also novel in that it uncovers the presence of two types of quartets. The visibility of the quartet resonances is acceptable, though sharper resonances would have been desirable to make the effect clearer.

In view of the novelty, timeliness and clarity of the work, I support its publication in Nature Communications. I have a couple of comments/questions for the authors:

1. Can the authors comment on the possible role of edge modes in the observed interference patterns, especially in Fig. 3. (e.g. in connection with <https://www.nature.com/articles/nnano.2017.24> or other instances in which edge modes - possibly trivial - arise in graphene structures (I am not one of the authors of that work))
2. Regarding the discussion of Floquet spectrum, and in particular Fig. 4a: can the authors distinguish the proposed mechanism from the more mundane Fiske resonances - i.e. ac-Josephson effect-like resonances due to noise amplification in the circuit/sample environment at microwave resonance frequencies of the environment?

Reviewer #2:

Remarks to the Author:

The authors describe measurements of the interference effects of Cooper Quartets, observed in a multi-terminal graphene Josephson junction where two terminals are tied by a flux loop. By biasing the superconducting contacts, the authors are able to identify a superconducting branch along the $V_1+V_2=0$ bias direction which they attribute to Cooper Quartet currents. Measuring this superconducting branch while varying the magnetic field, they are able to interfere the electrons mediating the supercurrent in the graphene resulting in both Φ and $\Phi/2$ periodicities. The latter they claim as being direct evidence of $4e$ charge transport, while the former they attribute to interference of entangled pairs of Cooper pairs generated in either of the flux loop contacts. Furthermore, by varying bias, the authors observe a $n=0$ transition in the superconducting phase of the current, which they explain as the evolution of a Landau-Zener transition between bands of the Floquet Andreev spectrum.

Engineering the Andreev spectrum of these multi-terminal Junctions is emerging as a promising route to explore nontrivial topological phenomena. To this end, the difficulty of making and measuring these devices has hindered progress substantially. The observation of a $\Phi/2$ periodicity directly demonstrates the $4e$ charge of these multiplet resonance lines. The observation of $4e$ coherent transport has substantial merit for applications in quantum information and on its own warrants publication.

The strong experimental evidence for the $4e$ charge transport aside, I have several points of concern regarding the split quartet and Floquet Andreev spectrum description of the $n=0$ transition that need to be addressed:

While I believe the split quartet process is possible given that the higher order nonlocal MAR are clearly demonstrated in the device, I find it interesting that the strength of the Fourier component associated with $2e$ transport is seemingly constant throughout the gatesweep of Figure 3. How many periods were used in the FFT? Is there also the possibility of a $2e$ process along the quartet biasing condition?

The device is measured to be ~ 2 Ohms, meaning there are potentially thousands of modes. How robust is the theory associated with creating the Floquet bands with so many modes to scatter into?

The quartet current seemingly disappears at the highest backgate voltage (40V) presented in the supplementary information. Is the quartet current visibility wrecked by the enhanced conductance (as in its there, but the coloring scheme doesn't show it) or does the addition of more modes really destroy the current? If it is the latter, is there a simple explanation for this?

In Figure 4, the applied voltage in the ballpark of $7\mu\text{V}$ places the measurement in the superconducting region. Why is there a residual resistance here at all?

Additionally, in this region, one has an abundance of Andreev processes, including typical $2e$ processes from a standard supercurrent. Here, tuning of a non-equilibrium voltage will effectively tune the Fermi level of one's Andreev spectrum. Given the large number of modes and thus Andreev bound states I don't think it is unreasonable that in this regime one starts cancelling components of your Andreev spectrum, which can yield a phase flip transition. Additionally, one would maybe expect to see the feature where the supercurrent vanishes in the bias-bias map. It may be explained that this feature only happens at the quartet bias condition, in this case it would be necessary to show there isn't a $n=0$ transition with incommensurate voltages.

Overall, while I believe that the underlying theory associated with this section is possible but probably completely obscured by the physics ignored in this region. The evidence is not convincing and the language surrounding the observation is far too strong unless the authors can provide additional information to confirm the Floquet picture to be 100% correct.

Minor points:

In the supplementary the authors state "together with signals from MARs in the background, is consistent with a quite transparent graphene junction". While it is true that dissipative local MAR between any pair of superconducting contacts in the device would only be visible if there are few or no losses in the graphene channel, they should only be observed in the instance of lower contact transparency. This is why the features fade out at higher backgate away from the Dirac peak in Figure 1. See for example, Figure 9 of Phys. Rev. B 99, 075416 (2019) for the expected visibility of these features in multi-terminal devices as a function of transparency, or alternatively Phys. Rev. B 64 144504 for the underlying theory of MAR in two terminal devices (Figure 16 being particularly relevant).

The backgate voltage of Figure 2b is never clearly stated.

The color scale of the differential maps probably isn't optimal for displaying the information used.

The applied voltage bias in the caption of Figure 3 where the cuts are taken is claimed to be 4V. Did the authors mean 0.4 mV (which would be consistent with 4V applied onto the e_4 divider, and the supplementary)?

Figure S5 left panel x-axis should be magnetic field, not voltage.

Reviewer #3:

Review of Ko-Fan Huang et al. "Interference of Cooper quartets in a multi-terminal graphene-based Josephson junction" for Nature Communication

This paper aims to establish more solid evidence for the existence of Cooper pair quartets than Ref. 8 (Pfeffer et al. PRB 2014) and Ref. 9 (Cohen et al. PNAS2018). The main argument in favor of quartets is made by interpreting an oscillating-in-flux differential resistance in a 4-lead device with a superconducting loop.

I have some concerns about the present manuscript, both in terms of the discussion of the result in the context of previous work and in terms of the technical implementation of the theoretical "Quartet" proposal. Before this paper can be published (in any journal, really), those concerns should be addressed.

In terms of the previous work, evidence for a phase-coherent Andreev reflection (AR) and the corresponding Andreev bound states (ABS) in multi-terminal Josephson junctions was already reported in Ref. 11 (Pankratova et al. PRX 2020). In fact, that paper describes the same device (4 terminals + SQUID loop) and the same measurement setup as used in this work. The semiconductor is some III-V structure instead of graphene, but that's probably a detail. Therefore, it seems obligatory to compare the present results to those already published in Ref. 11. I started exploring the comparison, and found a few interesting facts that certainly cannot be swept under the rug.

As far as I could see, both experiments aim to establish the existence of coherent AR from multiple superconducting leads. In Ref 11, the claim for coherent ABS is made by analyzing the modulation of the multi-terminal supercurrent with flux through the SQUID loop (see Ref. 11, Fig. 6, in particular). In that case the voltage drop across all the junction leads is zero. Finite-voltage measurement is also shown in Ref. 11. They show a forest of conventional two-terminal multiple Andreev reflection resonances in differential resistance. Yet, there is no "Quartet" resonance at $V_1 = -V_2$. On the other hand, in the present work there are no remarks about the modulation of the multi-terminal supercurrent (although it is clear from Fig. 2 that the authors were in a position to measure it) with flux, and there are no MAR resonances either. Yet the present authors claim the "Quartets", which implies the existence of coherent AR and ABS, a central claim of Ref 11.

So unless I am missing something, there is a bit of a controversy. One paper (Ref 11) reports coherent multi-terminal AR from supercurrent measurement and no effect at a finite bias. The present manuscript reports evidence of coherent multi-terminal AR from finite-voltage data but not from the supercurrent measurement.

If the present authors indeed reproduce the ground state data from Ref 11 AND also see the Quartet feature, then one can safely claim an interesting advance in the understanding of MTJJ. In particular, it would be interesting to understand why the Quartet feature appears in a graphene device but not in the other devices of Ref 11. On the other hand, if the authors cannot reproduce the data from Fig. 6 in Ref 11, then the claim for a Quartet is questionable, in my opinion.

Now I am going to make a few purely technical remarks that may undermine the author's Quartet interpretation.

(1) The "Quartet" anomaly appears in Fig. 2 when the voltage across the junctions is larger than the gap. For a 80 nm Al with a Ti sticking layer at 300 mK, I'd estimate the gap to be at 150 μ V at most, so $2\Delta \sim 300$ μ V. This is a major concern for the interpretation in terms of Floque bands. The junction between 1-2 gets a voltage above 0.5 mV and that's certainly way over the 2Δ .

Coherent dynamics of ABS should only be expected when the voltage across EVERY junction is lower (if not much lower) than the gap. This was nicely illustrated, for instance, by the study of Houzet, Meyer, Nazarov ... (see Eriksson et al. PRB 95, 075417 (2017)). The moment one of the junctions sees voltage $> 2\Delta$, there is a direct quasiparticle current in the system and the whole Josephson network becomes a mess. For example, coherent manifestations of the regular AC-Josephson effect, such as Shapiro steps, occur only for voltage below the gap.

Therefore, I would feel uneasy if the authors cannot reproduce the "Quartet" anomaly at a voltage bias of < 100 mV. I do not see a reason why this would not be possible. It seems to be a matter of reducing the bias resistor in the setup.

(2) You state that a cross-Andreev reflection is expected because the device leads have a size of about 1 μ m and so is the coherence length in the Al. However, 1 μ m is the coherence length in a crystalline Al. In an evaporated or sputtered film, the coherence length drops to about 100 nm.

Therefore, I would expect a very small amplitude for a cross-Andreev reflection in your geometry.

To mitigate my concern, have you tried repeating this experiment on a device with more narrow leads (say 0.5 μm instead of 1.5 μm)? That should lead to a stronger effect.

(3) In general, one should not be surprised that there is a flux-periodic resistance in a voltage biased Josephson network where one of the Josephson couplings is flux-tunable. To give you a simple example, consider a sub-gap current in a voltage-biased SQUID. This sub-gap current is due to the rectification of the Josephson oscillations, which depends on the external impedance, INCLUDING the Josephson inductance of the junction itself, which is flux-tunable. At the energy above the gap, the rectified current is messed up by the direct quasiparticle current, but some phase-modulation survives there as well. See for example Fig. 6.3 in the dissertation of J. Teufel (Yale University, 2008, Chapter 6).

In the present case, the network is a bit more complicated than a SQUID, there are four junctions coupled in a "ring" geometry, and one of them is phase-biased. A finite voltage of 0.3 mV produces oscillations at ~ 100 GHz, and at that frequency the loop impedance cannot be neglected, which means the terminals 0a and 0b are not short-circuited. This is how you get phase-sensitivity of the rectified Josephson current in other terminals of the network. Obviously, this is a major crack in the author's approach to establishing coherence and entanglement of Cooper pairing.

(4) The $F0/2$ -periodicity appears only for some range of gate voltages, around 25V. What is special about the device at this bias voltage?

Also, in the supplementary, the quartet thing is most prominent at -10V. How to reconcile these two observations? Have you tried measuring other devices?

(5) I noticed that you refer to the feature at $V1 = -V2$ as "supercurrent", but in reality the differential resistance is finite, and not only due to the bias resistor. Reduction of resistance might be a hint that some superconducting correlations take place, but I would not go as far as to claim the observation of Quartet supercurrent. Perhaps the language throughout the paper can be softened when the main data features are described.

(6) In the theory section, there is a voltage scale of $Vq = 7$ μV . However, the temperature is 300 mK $\gg 7$ μV . How is it possible that this energy scale is relevant? Was temperature taken into account in the Landau-Zener simulations? How many Andreev bands do you consider? In the picture I see one, but is this justified? Usually the number of bands equals the number of transparent channels. What experimental evidence can you provide in favor of 1 transparent channel?

In summary, I feel uneasy about the direct link between resistance oscillations and the transport of entangled Cooper pairs or the formation of the Floque Andreev bands. This link is most definitely indirect. I hope my specific remarks motivate my concerns and would help the authors to improve both the experimental study and the manuscript presentation. To be entirely honest, I would prefer to see a higher standard for the interpretation of a mesoscopic superconductivity experiment. This field has already suffered a great deal from the premature Majorana claims.

Response to Reviewers Comments

Reviewer #1:

The manuscript by Huang, Ronen et al. presents a timely and intriguing work on multi-terminal Josephson junctions (MT-JJs), a system that is receiving increasing attention due to the potential for realizing topological ABS spectra. The focus of this work is not on topology, but on Cooper pair entanglement (quartets) that can arise in MT-JJs. The work is very clear and although MT-JJs have been studied in graphene and other systems, there work is also novel in that it uncovers the presence of two types of quartets. The visibility of the quartet resonances is acceptable, though sharper resonances would have been desirable to make the effect clearer.

In view of the novelty, timeliness and clarity of the work, I support its publication in Nature Communications. I have a couple of comments/questions for the authors:

R1.0 We thank Reviewer#1 for the comments, and we look forward to sharing our results with the community.

1. Can the authors comment on the possible role of edge modes in the observed interference patterns, especially in Fig. 3. (e.g. in connection with <https://www.nature.com/articles/nnano.2017.24> or other instances in which edge modes - possibly trivial - arise in graphene structures (I am not one of the authors of that work))

R.1.1 We thank the reviewer for highlighting this paper to us. Although the physics in this study is interesting, it is not directly relevant to our work since we do not etch the graphene in any step of our fabrication process. Our multi-terminal Josephson junction is made by surface contacts to the bulk of graphene while the physical edges of graphene are far away from the JJs. We did not observe any conductance quantization in our measurements.

In the revision, we have updated the fabrication section in our manuscript to clarify the device structures associated with the reference above: “The graphene layer we use in our device is larger in size compared to the MT-JJ, and have not been etched in any step of the fabrication; thereby eliminating natural/etched edge effects from interfering with the JJ transport characteristics.”

2. Regarding the discussion of Floquet spectrum, and in particular Fig. 4a: can the authors distinguish the proposed mechanism from the more mundane Fiske resonances - i.e. ac-Josephson effect-like resonances due to noise amplification in the circuit/sample environment at microwave resonance frequencies of the environment?

R.1.2 We thank the Reviewer for this suggestion. We believe that Fiske steps are not involved here. They would take place as a resonance between the ac current and cavity modes at the junctions. Fiske steps in a JJ are modulated by a magnetic field corresponding to the Fraunhofer modulation of the Josephson critical current. On the contrary, the quartet V-I anomalies we observe are modulated by a much lower magnetic field corresponding to flux quanta entering the loop external to the junction, not the junction itself. Moreover, Fiske steps would occur at any junction in our multiterminal structure as soon as V_1 or V_2 matches a cavity frequency, which was not observed in our experiment. The anomaly we study occurs only along the $V_1=-V_2$ line.

In the revision, we made a brief comment on Fiske steps for clarification: “Another possible cause of a voltage-dependent anomaly are Fiske steps [Ref 35], realized in a tunnel junction that is coupled to a cavity resonance. To confine these modes, a sufficiently large magnetic field is required, corresponding to the area of the JJ. In our study, the anomaly appears in a much lower field range, corresponding to the larger SC loop area. Therefore, our observation cannot be associated with the Fiske steps.”

Reviewer #2:

The authors describe measurements of the interference effects of Cooper Quartets, observed in a multi-terminal graphene Josephson junction where two terminals are tied by a flux loop. By biasing the superconducting contacts, the authors are able to identify a superconducting branch along the $V_1+V_2=0$ bias direction which they attribute to Cooper Quartet currents. Measuring this superconducting branch while varying the magnetic field, they are able to interfere the electrons mediating the supercurrent in the graphene resulting in both Φ and $\Phi/2$ periodicities. The latter they claim as being direct evidence of $4e$ charge transport, while the former they attribute to interference of entangled pairs of Cooper pairs generated in either of the flux loop contacts. Furthermore, by varying bias, the authors observe a π -0 transition in the superconducting phase of the current, which they explain as the evolution of a Landau-Zener transition between bands of the Floquet Andreev spectrum.

R.2.0 We thank Reviewer#2, summarizing well our experimental observations. However, we wish to point out that by varying the bias, we observe a π -0 phase transition of quartet current's flux dependence, but not of the phase of quartet current itself. Measuring the phase of the quartet current would require a much more complex setup, which is beyond the scope of this work.

Engineering the Andreev spectrum of these multi-terminal Junctions is emerging as a promising route to explore nontrivial topological phenomena. To this end, the difficulty of making and measuring these devices has hindered progress substantially. The observation of a $\Phi/2$ periodicity directly demonstrates the $4e$ charge of these multiplet resonance lines. The observation of $4e$ coherent transport has substantial merit for applications in quantum information and on its own warrants publication.

R.2.1 We very much appreciate Reviewer#2's insights. While the existence of quartets indeed has been implied in previous experiments, this manuscript reports the first observation of $\Phi/2$ periodicity, suggesting the coherent transport of entangled Cooper pairs.

In the revision, to emphasize the experimental observation of $4e$ charge periodicity, we modify the title: **"Evidence for $4e$ periodicity of Cooper quartets in a biased multi-terminal graphene-based Josephson junction"**

The strong experimental evidence for the $4e$ charge transport aside, I have several points of concern regarding the split quartet and Floquet Andreev spectrum description of the π -0 transition that need to be addressed: While I believe the split quartet process is possible given that the higher order nonlocal MAR are clearly demonstrated in the device, I find it interesting that the strength of the Fourier component associated with $2e$ transport is seemingly constant throughout the gate sweep of Figure 3. How many periods were used in the FFT? Is there also the possibility of a $2e$ process along the quartet biasing condition?

R.2.2 The reviewer correctly pointed out that the Fourier component associated with Φ periodicity stays constant, while the $\Phi/2$ varies sensitively with the gate voltage. All quartet processes, split or not, involve two direct and two nonlocal AR. Unfortunately, these higher order nonlocal ARs involve more complex processes. Therefore, there is no simple and intuitive explanation for the relative weights of the two Fourier components. Moreover, the gate voltage not only alters the number of channels in graphene, but also modulates the Josephson junction transparency (see **R.2.4**), making it even more challenging for a more systematic analysis. For the FFT in Fig.3, we use ~ 4 -8 periods of oscillation presented in Fig. 3a-b with ~ 60 data points and the standard discrete Fourier transform algorithm available in MATLAB, ensuring that the FFT spectrum represent the periodic modulations of the signal. We also wish to note that our junction is biased in the range of 30 times smaller than

the superconducting energy gap (see **R2.8**). Thus, it is unlikely the DC $2e$ process and MARs can be effective in the quartet bias condition we explore in Fig.3.

In the revision, following suggestions from the reviewer, we have added: “We observe that the Fourier component associated with Φ periodicity stays constant while the $\Phi/2$ component varies sensitively with the gate voltage. Although a full understanding of this dependence is beyond the scope of this work, it indicates that the strengths of the two different (i.e., conventional and split) quartet processes are determined by the relative contact couplings, which are tunable via gating (see Section 5 in SI).”

We also have clarified this point in the captions of figure 3a-b: “Right panel discrete Fourier transform (DFT) analysis of the data shows prominent harmonics. The quartet is biased at $V_1=-V_2=0.4$ mV, where the DC $2e$ process and MARs are not effective.”

The device is measured to be ~ 2 Ohms, meaning there are potentially thousands of modes. How robust is the theory associated with creating the Floquet bands with so many modes to scatter into?

R.2.3 The Reviewer is correct that there are many modes in the graphene channel, and thus a single mode dot model presented in the text is a rather simplified picture. We made our effort to extend this single mode model to two-channel dot (SI S5Be). Also, recently we extend the model to a multi-channel problem (arXiv: 2110.09870). In this new extended model, we have found two re-occurring key features for multi-level models in comparing the single level dot model to the multi-channel junction: (i) non-monotonic voltage dependence of the anomaly which occurs due to resonances induced within the Andreev spectrum and (ii) the flux modulation of these resonances is due to modulation of the junction parameters. Both features commonly appear both in the single level dot model and the multi-channel junction, as shown in numerical studies (SI and arXiv: 2110.09870). Furthermore, there is no qualitative difference between the "inversion" found in a single and in a multi-channel model. However, for a multi-channel device, an intuitive physical picture for Floquet physics becomes more involved as there are many bands in the system. In this regard, the Floquet bands picture we used in Fig. 4 should be considered only for a simplified qualitative description. Full quantitative consideration of non-equilibrium dynamics of multi-terminal can be more involved and is beyond the scope of our current work. We also found that there is a mistake in the scale of conductance data in the original figure shown in Fig. 1-2. The normal state resistance measured in quartet condition is ~ 80 Ohm, which correspond to 10-100 modes in the graphene channel.

In the revision, we have clarified this point: “This picture is corroborated by detailed non-equilibrium calculations within a single-level quantum dot model (see the SI S5Ba). While this simple model is not intended to be quantitative in a multi-channel junction as the one in our experiment, similar physics can be applied: when the Josephson frequency due to V_q matches the spacing between the ABS levels, resonance would occur, resulting in an oscillation of I_{qc} with V_q . Numerical studies on multi-channel models have confirmed this picture (see SI S5Be).” In addition, we also converted the scales for G_1 and G_2 in the figures. We now present the normalized conductance G_1/G_N and G_2/G_N , where the normal state resistance $G_N^{-1} = 80 \Omega$ is measured separately from a 4-terminal measurement (see the SI).

The quartet current seemingly disappears at the highest backgate voltage (40V) presented in the supplementary information. Is the quartet current visibility wrecked by the enhanced conductance (as in its there, but the coloring scheme doesn't show it) or does the addition of more modes really destroy the current? If it is the latter, is there a simple explanation for this?

R.2.4 We thank the reviewer for raising this point. We have indeed shown a zoomed-in region of the back-gate voltage in fig3a-b as well as a zoomed-out region in fig S4. As a function of back gate, we observe non-monotonic

intensity of the quartet signal. This behavior arises from the change in contact transparencies when the back-gate voltage is adjusted. This also complicates the roles that the number of channels plays in the experiment.

In the revision, we have clarified this point: “The relative strength of the periodicities is tuned non-monotonically, since V_{bg} modifies the number of channels in graphene as well as the coupling of S-electrodes, which modifies the ABS spectrum.”

In Figure 4, the applied voltage in the ballpark of $7\mu\text{V}$ places the measurement in the superconducting region. Why is there a residual resistance here at all? Additionally, in this region, one has an abundance of Andreev processes, including typical $2e$ processes from a standard supercurrent.

R.2.5 The reviewer raises a good point which can be understood by considering the difference between a 2-terminal (2T)-JJ and a quartet MT-JJ. In a 2T-JJ, the DC supercurrent necessitates a zero-voltage difference between the two SC terminals. In a biased 2T-JJ, the DC supercurrent vanishes, leaving only the AC supercurrent. In a MT-JJ, at the quartet condition, where the two junctions are biased asymmetrically, we are beyond the 2T-DC critical current limit. In this regime, a quasi-particle current can flow in parallel to the DC quartet supercurrent. Thus, unlike the 2T-JJ case, the two mechanisms are simultaneously transmitting charges as depicted by the background conductance in Fig. S3c. Therefore, in the biased quartet conduction regime there is no standard $2e$ DC supercurrent, which is distinctively different from the superconducting region.

In the revision, we have clarified this point: “Furthermore, as opposed to the 2-terminal case, in a MT-JJ, quasi-particle current and quartet DC current flow simultaneously due to the inequivalent chemical potential of the SC contacts under the quartet condition. We delineate our signal from the quasi-particle current contribution by measuring an oscillatory differential conductance, following the $I_{qc}(\Phi)$ variation along the quartet bias condition ($V_1 = -V_2$) on top of the quasi-particle current background (see Fig. S3c).”

Here, tuning of a non-equilibrium voltage will effectively tune the Fermi level of one’s Andreev spectrum. Given the large number of modes and thus Andreev bound states I don’t think it is unreasonable that in this regime one starts cancelling components of your Andreev spectrum, which can yield a phase flip transition. Additionally, one would maybe expect to see the feature where the supercurrent vanishes in the bias-bias map. It may be explained that this feature only happens at the quartet bias condition, in this case it would be necessary to show there isn’t a π -0 transition with incommensurate voltages.

R.2.6 We thank the reviewer’s suggestion of an alternative scenario. Indeed, in a three-terminal setup where a third *normal* contact is sourcing normal current into the JJ, a 0- π transition can be triggered (for example, Baselmans et al., Nature 397, 43 (1999)). However, we wish to point out that our junctions are all made of SC contacts, and the bias is much less than the SC gap. Thus, there is no normal current injection in our experiment. Furthermore, the 0- π transition of the magnetic flux phase threading through the loop shown in Fig. 4 is strictly for the quartet bias condition ($V_1 = -V_2$). This 0- π transition feature

Fig. R1 **A.** G_2/G_N as a function of magnetic fields and V_1 at fixed $V_2=+7.5\mu\text{V}$. **B.** Traces obtained from **A**, along fixed V_1 to demonstrate 0- π phase change only occurs near $V_1=-V_2$.

does not appear at incommensurate bias voltage configuration ($V_1 \neq -V_2$) as shown in Fig. R1 above.

In the revision, following the reviewer's suggestion, we stated "We note that the quartet bias condition ($V_1=-V_2$) is essential for observing this $0-\pi$ phase change as such phase change is absent at incommensurate bias condition ($V_1 \neq -V_2$) (see Fig. S7 in SI).", showing Fig. R1 as Fig. S7 in SI.

Overall, while I believe that the underlying theory associated with this section is possible but probably completely obscured by the physics ignored in this region. The evidence is not convincing and the language surrounding the observation is far too strong unless the authors can provide additional information to confirm the Floquet picture to be 100% correct.

R.2.7 We thank the reviewer for the suggestions. As we discussed in R.2.3., the presence of multi-channels in our MT-JJs makes a direct application of the Floquet theory. But for simplistic, the theory we provide is based on a single mode. In this regard, we agree with the reviewer that it is necessary to clarify the limitation of the model used in our analysis.

In the revision, following the suggestions of the reviewer, we have revised several of our claims regarding the connection between the Floquet picture and the interpretation of our experimental findings. In particular, we now state explicitly that we use a Floquet picture to illustrate the more general concept of a self-induced resonance, which requires a model with few-modes: "This demonstrates, by analogy to other microwave resonance phenomena in metallic junctions, that a non-monotonic dependence of the quartet current on applied voltages is expected. In a set of non-equilibrium ABSs, its effective separation depends on the bias voltage in analogy to the Floquet bands (39-40) separated by $2eV_q$, emerging from the periodically driven Bloch bands (34,41).

This picture is corroborated by detailed non-equilibrium calculations within a single-level quantum dot model (see the SI S5Ba). While this simple model is not intended to be quantitative in a multi-channel junction as the one in our experiment, similar physics can be applied: when the Josephson frequency due to V_q matches the spacing between the ABS levels, resonance would occur, resulting in an oscillation of I_{qc} with V_q . Numerical studies on multi-channel models have confirmed this picture (see SI S5Be)."

Minor points:

In the supplementary the authors state "together with signals from MARs in the background, is consistent with a quite transparent graphene junction". While it is true that dissipative local MAR between any pair of superconducting contacts in the device would only be visible if there are few or no losses in the graphene channel, they should only be observed in the instance of lower contact transparency. This is why the features fade out at higher backgate away from the Dirac peak in Figure 1. See for example, Figure 9 of Phys. Rev. B 99, 075416 (2019) for the expected visibility of these features in multi-terminal devices as a function of transparency, or alternatively Phys. Rev. B 64 144504 for the underlying theory of MAR in two terminal devices (Figure 16 being particularly relevant).

R.2.8 We thank the Reviewer for commenting on the "MARs background" in the SI. In the original submission, we casually used "MARs background", referring to quasi-particle current contribution at small but finite bias across the JJs. The bias voltage applied here is about 30 times smaller than the superconducting gap, which corresponds to extremely high order MAR process. As the reviewer pointed out correctly, such high order MARs require extremely clean samples with low contact transparency. Thus, the experimental observation in our MT-JJ is unlikely MARs, and should better be termed as a quasi-particle background current.

In the revision, we replace the “MARs background” with “a quasi-particle background current” and have added: “One should also add that MAR features might fade out at higher back-gate voltages due to an increase transmission”, as well as added the citations the Reviewer was referring to.

The backgate voltage of Figure 2b is never clearly stated.

We have added this piece of information in the main text and in the Figure caption.

The color scale of the differential maps probably isn't optimal for displaying the information used.

Color scale has been changed.

The applied voltage bias in the caption of Figure 3 where the cuts are taken is claimed to be 4V. Did the authors mean 0.4 mV (which would be consistent with 4V applied onto the e4 divider, and the supplementary)?

The reviewer is correct.

We corrected this mistake in the revision.

Figure S5 left panel x-axis should be magnetic field, not voltage.

In S5, we show the relation between quartet conductance (G_1) and quartet critical voltage (V_{qc}). In the left panel, the bias voltage is fixed while the five datapoints are taken under five different values of magnetic flux.

We have added a legend to S5 in the revision.

Reviewer #3:

This paper aims to establish more solid evidence for the existence of Cooper pair quartets than Ref. 8 (Pfeffer et al. PRB 2014) and Ref. 9 (Cohen et al. PNAS2018). The main argument in favor of quartets is made by interpreting an oscillating-in-flux differential resistance in a 4-lead device with a superconducting loop.

I have some concerns about the present manuscript, both in terms of the discussion of the result in the context of previous work and in terms of the technical implementation of the theoretical “Quartet” proposal. Before this paper can be published (in any journal, really), those concerns should be addressed.

R.3.0 We thank Reviewer#3 for careful reading of our manuscript and providing constructive suggestions. As in the following discussion, we hope that we have clarified and improved the manuscript to address all the reviewer’s concerns.

In terms of the previous work, evidence for a phase-coherent Andreev reflection (AR) and the corresponding Andreev bound states (ABS) in multi-terminal Josephson junctions was already reported in Ref. 11 (Pankratova et al. PRX 2020). In fact, that paper describes the same device (4 terminals + SQUID loop) and the same measurement setup as used in this work. The semiconductor is some III-V structure instead of graphene, but that’s probably a detail. Therefore, it seems obligatory to compare the present results to those already published in Ref. 11. I started exploring the comparison, and found a few interesting facts that certainly cannot be swept under the rug.

R.3.1 The reviewer correctly pointed out that a similar experimental approach has recently been pursued in Ref 11, employing Al-InAs based MT-JJs. We believe our work is complementary rather than controversial to Ref 11. As we will clarify more details in the later responses, in our study we obtained transport features at bias voltages much smaller than the energy gap, including quartet supercurrent at asymmetric bias condition. On the other hand, the InAs semiconductor-based MT-JJ studied in Ref 11 provided fewer modes in the semiconductor channel with lower disorder than our graphene MT-JJs. This condition yields a larger $I_c R_N$ product in the semiconductor MT-JJ, signaled by the observation of higher order MARs at relatively larger bias voltages. We will make more quantitative comparison in responses (R.3.2 and R.3.3).

In revision, following the reviewer’s suggestion, we have made considerable effort to contrast the difference between our work in and Ref 11. **Specifically, we have added to a new section in SI: S6 provides the critical current contour oscillation as a function of the flux threading through the loop and can be directly compared to the work presented in Ref 11.**

As far as I could see, both experiments aim to establish the existence of coherent AR from multiple superconducting leads. In Ref 11, the claim for coherent ABS is made by analyzing the modulation of the multi-terminal supercurrent with flux through the SQUID loop (see Ref. 11, Fig. 6, in particular). In that case the voltage drop across all the junction leads is zero. Finite-voltage measurement is also shown in Ref. 11. They show a forest of conventional two-terminal multiple Andreev reflection resonances in differential resistance. Yet, there is no “Quartet” resonance at $V_1 = -V_2$.

R.3.2 As the reviewer summarized well, the main experimental findings in Ref 11 are (a) observation of magnetic field dependence of the critical current contour (CCC) in the 2d and 3d current space; and (b) a strong set of MAR line families, corresponding to higher order (up to $n=10$) MARs occurring between any pair of terminals. Interestingly, however, (c) absence of any special features in the quartet resonance condition at $V_1=-V_2$.

Comparing these main results, we find that our graphene MT-JJs exhibit indeed (a), the magnetic field dependent CCC as shown in Fig. 2b. We also notice that our device exhibits (b), MAR line families, especially near the charge neutrality regime in gate voltages, where MARs up to $n=4$ can be resolved (see Fig. R2). It is important to note that the bias voltage shown in Fig. 1e and Fig.2b are applied bias voltage including the resistors from RC filters that are connected to the device in series. Actual bias voltage applied on JJs thus depends on both R_{RC} and $R_N=G_N^{-1}$, G_N^{-1} is back-gate dependent and which has maximum near the Dirac point of the graphene channel. We find that given the bias voltage V , the junction bias voltage V_J beyond the supercurrent regime is $V_J=VR_N/[R_N+2R_{RC}]$. Thus, V_J becomes smaller as the gate voltage is tuned away from the Dirac point. We can see this from Fig. 1e in our revision. Near the Dirac point ($V_{bg} = -35$ V), where V_J is in the range of ± 0.15 mV and considered to be a large bias ($R_{RC}=200 \Omega$ and $R_N=80 \Omega$), we can resolve MARs up to $n=5$. As V_{bg} increases, R_N becomes smaller and V_J becomes a smaller bias even with the same applied bias V . This effect can be seen in that the MARs family lines move out of our bias window as the back-gate voltage is increased (Fig. 1e). At the back-gate voltages ($V_{bg} > 25$ V) where we measured the quartet current, V_J is estimated to be $< 10 \mu\text{V}$. In this regime, the ratio between the observed quartet voltage to the SC gap is 1 to 30. Therefore, to observe MARs in the region where we observe quartets, one would need to go to orders as high as 20-30. This is unlikely to achieve in such a system where transport is not ballistic. And indeed, MARs are not observed at these regions in our devices. This is a huge difference between our work and Ref. 11. Notice that in the three existing experiments where quartets have been observed (with diffusive metallic, with nanowire and with graphene junctions), the voltage scales are all comparable (**5-10 μV**), which are much smaller than the AI gap. Quartet ABSs only require four ARs (irrespective of the V_J value), while MARs require a number that scale as $1/V_J$. Thus, we speculate that the reason that quartets are not seen in Ref. 11 could be the following: at higher voltages, MAR lines, which involve pairs of terminals but not three of them simultaneously like quartets, are too bright and mask any possible quartet. After all, in the process of creating quartets, two of the ARs are crossed ARs, which suffer from a geometrical factor. In addition, due to the frequency dependency, the quartet signal decrease as voltage increases. Thus, MARs and quartets are competing processes, and we claim that MARs are easier to be observed at high voltages and quartets at low voltages.

In the revision, we made comparison of our results to that of Ref. 11 by adding: **“In addition to the two-terminal Josephson currents (i)-(iii), we observe another supercurrent signal along the $V_1 = -V_2$ line, as can be seen in Fig 2b. This line originates from the sharp black lines, which define the 2-terminal critical current contour (CCC). To ascertain the intrinsic nature of this signal and that it is due to quartets, let us first remark that no clear multiple Andreev reflections (MARs) are observed in this sample in the bias voltage range where we observe a quartet signal. Indeed, given the low value of the junction voltage, those MARs, whether local or nonlocal, would have very high orders. In a non-ballistic graphene with interface scattering, such high-order MARs are unlikely to take place, in contrast to clean InAs 2DEG samples such as those in Ref. 11. In the work of Ref.11, where the**

Fig. R2: Normalized gate dependence of the supercurrent, dI/dV as a function of the bias voltage and global back-gate voltage V_{bg} . The critical current reaches the minimum as graphene is tuned to the Dirac point near $V_{bg} = -32$ V. A cut at $V_{bg} = -32$ V shows MARs up to the 5th order. Brown line illustrates the MAR evolution as a function of gate voltage due to the decrease in the normal resistance of the junction.

critical currents are high, the situation is very different: several bright local MAR lines were observed, but no supercurrent was observed along the $V_1 = -V_2$ line beyond the CCC. This is not surprising because quartets require four ARs, two local and two nonlocal processes, and are easily masked by bright local MARs. Notice that the same conditions (low voltage compared to the gap, no MARs or very weak ones) were met in Refs. 8-9 and a quartet line was indeed observed.”

On the other hand, in the present work there are no remarks about the modulation of the multi-terminal supercurrent (although it is clear from Fig. 2 that the authors were in a position to measure it) with flux, and there are no MAR resonances either. Yet the present authors claim the “Quartets”, which implies the existence of coherent AR and ABS, a central claim of Ref 11.

R.3.3 As a sanity check, we have our multi-terminal device operate as a two-terminal junction with a loop, which works as a SQUID. The magnetic field modulation of the critical current was studied (Figure 1c). In the three-terminal operation, we did not investigate the modulation of the CCC as extensively as in Ref. 11 since our focus is the quartet modulation. However, we do have data showing CCC modulation in the non-quartet bias condition as the magnetic flux is swept (we will discuss this in R.3.4 below). We also state above that the main difference between Ref.11 and our MT-JJ work lies in the measurement voltage bias regime. In Ref.11, the MARs appeared in relatively high bias, while in our work we concentrate our effort in the low bias regime as discussed R.3.1 above.

In the revision, we have added an additional y-axis scale to Figure 4a to clarify that the quartet configuration is achieved in a scenario where all contacts are in different chemical potentials. The additional y-axis in Figure 4a corresponds to the diagonal yellow dotted line in figure 2b. Fig4a caption was changed accordingly.

We have added in the revised text: “Both voltage scales (at the junction and at the circuit resistance) are presented, to make the correspondence with the quartet line in the zero-field map of Figure 2b. This shows that this voltage region lies beyond the Josephson regime (black line in the bottom of Figure 4a).”

So unless I am missing something, there is a bit of a controversy. One paper (Ref 11) reports coherent multi-terminal AR from supercurrent measurement and no effect at a finite bias. The present manuscript reports evidence of coherent multi-terminal AR from finite-voltage data but not from the supercurrent measurement. If the present authors indeed reproduce the ground state data from Ref 11 AND also see the Quartet feature, then one can safely claim an interesting advance in the understanding of MTJJ. In particular, it would be interesting to understand why the Quartet feature appears in a graphene device but not in the other devices of Ref 11. On the other hand, if the authors cannot reproduce the data from Fig. 6 in Ref 11, then the claim for a Quartet is questionable, in my opinion.

R.3.4 We wish to assure the reviewer that there is no controversy between our work and the work presented in Ref 11. The authors in Ref 11 used current bias for their MT-JJ. Typically, in this measurement configuration, the junction bias voltage can easily be raised to non-supercurrent regime due to strong non-linear I-V curves. Therefore, in this measurement, the bias voltage is large enough such that one can easily observe the MARs. But at the same time, it is beyond the low bias regime where one can observe the quartet signal (**R.3.2**). In our measurement scheme, we have the voltage bias across two serially connected resistors, $2R_{AC}$. It allows us to carefully control the junction bias voltages. As shown in Fig. 1e in the revision, we demonstrate that we can observe both MARs and quartet in our devices by controlling the normal state graphene channel resistance via back-gate. We also recognize the need the reviewer has raised to directly demonstrate CCC modulation by magnetic

Fig. R3. Differential conductance along the $V_1=V_2$ line, showing flux dependence oscillations of CCC.

flux. As shown in Fig. R3, we perform a similar measurement to the one shown in Fig.1 in the main text where we voltage bias both S1 and S2 while keeping S0a and S0b grounded. We measure the differential conductance as a function of bias as well as magnetic field. To clarify the measurement that would be a measurement along the S1-S2 supercurrent – line (iii) in Fig 2 of the main text. One can clearly see an oscillation of the critical current contour in similarity to what is seen in Ref 11. We believe this addition answers to the reviewer’s concern by showing the CCC oscillations as a function of flux in addition to the non-equilibrium observation in our device.

In the revision, we have added: “In Fig S6 we measure differential conductance of the multi-terminal JJ as a function of magnetic field and along the $V_1=V_2$ condition. In this measurement we are performing a cut along the $+45^\circ$ line of the critical current contour and observing its behavior as a function of magnetic field. This is an important point considering previous works on MT-JJ, specifically ref 11 in our main text.”

Now I am going to make a few purely technical remarks that may undermine the author’s Quartet interpretation.

(1) The “Quartet” anomaly appears in Fig. 2 when the voltage across the junctions is larger than the gap. For a 80 nm Al with a Ti sticking layer at 300 mK, I’d estimate the gap to be at 150 μ V at most, so $2\Delta \sim 300 \mu$ V. This is a major concern for the interpretation in terms of Floque bands. The junction between 1-2 get’s a voltage above 0.5 mV and that’s certainly way over the 2Δ .

R.3.5 We use a voltage bias scheme in our experiments (the complete circuit in Figure S1). The voltage indicated in the plots is applied on both the RC filters (with 200 Ohm resistance) and the junction itself. At the gate voltage where we measured the quartet signal, the graphene channel resistance is $< 80 \Omega$. Therefore, the actual voltage applied on the junction is at least ~ 5 times smaller than the voltage read in the plots. In our experiment, considering the non-linear I-V of the JJ, we also directly measure the junction voltage using voltage preamp. Fig. 4a in the revision now shows the measured quartet voltage V_q together with applied bias voltage. V_q is 100 times smaller than the bias voltage $V_1=-V_2$.

In the revision, to address the reviewer’s concern, we made the following changes:

- We added left axis in Fig. 4a for a direct comparison to the junction bias V_q and applied bias voltage in the circuit V_1 .
- We have added: “Voltages V_1 and V_2 are applied to the total circuit including R_{RC} , they are about 100 times larger than the actual voltage applied at the junction (see circuit in Fig S1).”

Coherent dynamics of ABS should only be expected when the voltage across EVERY junction is lower (if not much lower) than the gap. This was nicely illustrated, for instance, by the study of Houzet, Meyer, Nazarov ... (see Eriksson et al. PRB 95, 075417 (2017)). The moment one of the junctions sees voltage $> 2\Delta$, there is a direct quasiparticle current in the system and the whole Josephson network becomes a mess. For example, coherent manifestations of the regular AC-Josephson effect, such as Shapiro steps, occur only for voltage below the gap. Therefore, I would feel uneasy if the authors cannot reproduce the "Quartet" anomaly at a voltage bias of < 100 mV. I do not see a reason why this would not be possible. It seems to be a matter of reducing the bias resistor in the setup.

R.3.6 As the reviewer stated, indeed seeing a quartet signal at energies higher than the gap and not seeing the quartet signal at lower energies is not expected. We apologize again for the misunderstanding of the voltage ranges we have measured. All our observations are concentrated around $\sim 7\mu\text{V}$ which is indeed much smaller than the superconducting gap. One can also see that the quartet line is seen immediately after the CCC region and indeed it quickly decays as the reviewer predicts.

In the revision, **we address this issue as we discussed in R3.3 and R3.5.**

(2) You state that a cross-Andreev reflection is expected because the device leads have a size of about $1\mu\text{m}$ and so is the coherence length in the Al. However, $1\mu\text{m}$ is the coherence length in a crystalline Al. In an evaporated or sputtered film, the coherence length drops to about 100nm . Therefore, I would expect a very small amplitude for a cross-Andreev reflection in your geometry.

To mitigate my concern, have you tried repeating this experiment on a device with more narrow leads (say $0.5\mu\text{m}$ instead of $1.5\mu\text{m}$)? That should lead to a stronger effect.

R.3.7 We thank the reviewer for the comment. Indeed, in the quartet process two local Andreev reflection (AR) and two crossed Andreev reflection (CAR) processes occur. Usually when we refer to CAR, the coherence length of the Al is a crucial parameter since a CAR is occurring across a superconducting metal as in Nature Phys. 13, 693 (2017). In our case when we state CAR, it is a local AR occurring in one of the leads, but having the impinging electron and the reflected hole originate from two different leads. This process should not depend in first order on the length of the contact, since the AR is local instead of across a metal region. However, we find empirically that we do not see the quartet signals in devices with narrower leads $\sim 0.8\mu\text{m}$. We think the reason is that the wider the junction is, the more likely cross-Andreev process would occur as the three terminals are coupled on a length shorter than the coherence length, which is the case on our samples.

In the revision we have clarified the difference of the CAR term we have used, we have added: **"We note that the crossed Andreev reflection is a local AR at one of the SC electrodes as opposed to CAR across a SC metal."**

(3) In general, one should not be surprised that there is a flux-periodic resistance in a voltage biased Josephson network where one of the Josephson couplings is flux-tunable. To give you a simple example, consider a sub-gap current in a voltage-biased SQUID. This sub-gap current is due to the rectification of the Josephson oscillations, which depends on the external impedance, INCLUDING the Josephson inductance of the junction itself, which is flux-tunable. At the energy above the gap, the rectified current is messed up by the direct quasiparticle current, but some phase-modulation survives there as well. See for example Fig. 6.3 in the dissertation of J. Teufel (Yale University, 2008, Chapter 6).

In the present case, the network is a bit more complicated than a SQUID, there are four junctions coupled in a "ring" geometry, and one of them is phase-biased. A finite voltage of 0.3mV produces oscillations at $\sim 100\text{GHz}$,

and at that frequency the loop impedance cannot be neglected, which means the terminals 0a and 0b are not short-circuited. This is how you get phase-sensitivity of the rectified Josephson current in other terminals of the network. Obviously, this is a major crack in the author's approach to establishing coherence and entanglement of Cooper pairing.

R.3.8 We agree that it is not surprising that there is a periodic magnetic field modulation of the junction resistance on the diagonal "quartet" line. However, in addition to the modulation, we show a $0-\pi$ transition of the phase of this modulation, which is a new observation that has never been reported before in the quartet experiments.

We also appreciate the Reviewer's concern regarding the loop impedance with the AC Josephson effect. However, since we apply a voltage which is in the range of a few micro volts, the oscillations would be in the GHz range. We estimate the loop impedance at this frequency range is $2.5 \text{ m}\Omega$, which is negligible compared to the series resistance and the normal junction resistance.

In the revision we have provided the loop impedance estimation at the quartet bias condition: "Finally, loop impedance effects may also be neglected, at the measured voltage ranges ($\sim\mu\text{V}$, $\sim\text{GHz}$) as it is estimated at $2.5\text{m}\Omega$."

(4) The $F_0/2$ -periodicity appears only for some range of gate voltages, around 25V. What is special about the device at this bias voltage?

Also, in the supplementary, the quartet thing is most prominent at -10V. How to reconcile these two observations? Have you tried measuring other devices?

R.3.9 We thank the Reviewer in raising this point. We have indeed showed a zoomed-in region of the back gate voltage in Fig3a-b as well as a zoomed-out region in Fig S4. As a function of back gate, we observe non-monotonic intensity of the quartet signal. The change in the quartet signal may be due to the modification of (i) carrier density in the junction and (ii) the contact transparencies as the back-gate voltage is tuned. Since having disorder in this system makes the gate voltage dependence non-trivial, it is hard to expound the complex relation between quartet signal intensity and gate voltage. We have measured 3 devices and they all showed similar behavior.

In the revision, we have clarified this point: "The relative strength of the periodicities is tuned non-monotonically, since V_{bg} modifies the number of channels in graphene as well as the coupling of S-electrodes, thus modifying the ABS spectrum."

(5) I noticed that you refer to the feature at $V_1 = -V_2$ as "supercurrent", but in reality the differential resistance is finite, and not only due to the bias resistor. Reduction of resistance might be a hint that some superconducting correlations take place, but I would not go as far as to claim the observation of Quartet supercurrent. Perhaps the language throughout the paper can be softened when the main data features are described.

R.3.10 In the revision, we have softened our claims following the reviewer's suggestion. Most notably, we replace the word "quartet supercurrent" to "quartet current contribution." We also change the title of the paper to: "Evidence for $4e$ periodicity of Cooper quartets in a biased multi-terminal graphene-based Josephson junction", emphasizing more the experimental observation. We also softened the claim regarding the Floquet Andreev bound states as we pointed out the clear limitation of our approach (please see **R.2.3** for more details): "This picture is corroborated by detailed non-equilibrium calculations within a single-level quantum dot model (see the SI S5Ba). While this simple model is not intended to be quantitative in a multi-channel junction as in our experiment"

(6) In the theory section, there is a voltage scale of $V_q = 7 \text{ uV}$. However, the temperature is $300 \text{ mK} \gg 7 \text{ uV}$. How is it possible that this energy scale is relevant? Was temperature taken into account in the Landau-Zener simulations? How many Andreev bands do you consider? In the picture I see one, but is this justified? Usually the number of bands equals the number of transparent channels. What experimental evidence can you provide in favor of 1 transparent channel?

R.3.11 The theory is based on $T=0$. We did not draw a strict quantitative comparison between the theory and the experiment in this study, the role of the model is to provide intuition for the reader.

In the revised manuscript we have stressed the importance of the single mode picture due to its elegant explanation of the π -shift flux modulation of the quartet supercurrent. In comparing the single level dot model to the multichannel junction, we have found two re-occurring key features: (i) the non-monotonous voltage dependence of the anomaly, due to resonances it induces within the Andreev spectrum and (ii) the flux modulation of these resonances due to modulation of the junction parameters. Both features happen in the single level dot model as well as the multichannel junction, as shown by numerical studies (SI): there is no qualitative difference between the "inversion" found in a single and a multichannel model, only in the last case, finding an intuitive physical picture is more difficult, as Floquet physics becomes more involved in a many-band system.

In the revision, we have clarified this point: "This picture is corroborated by detailed non-equilibrium calculations within a single-level quantum dot model (see the SI). It is not intended to be quantitative in a many-channel junction as ours, but the same physics applied: as soon as the Josephson frequency due to V_q matches the spacing between some ABS levels, resonance occur together with an oscillation of I_{qc} with V_q . Numerical studies on many-channel models fully confirm this picture (see SI)."

In summary, I feel uneasy about the direct link between resistance oscillations and the transport of entangled Cooper pairs or the formation of the Floquet Andreev bands. This link is most definitely indirect. I hope my specific remarks motivate my concerns and would help the authors to improve both the experimental study and the manuscript presentation. To be entirely honest, I would prefer to see a higher standard for the interpretation of a mesoscopic superconductivity experiment. This field has already suffered a great deal from the premature Majorana claims.

R.3.12 We thank the reviewer for constructive suggestions. As we discussed above, we improved the manuscript following the reviewer's suggestions. We hope the reviewer can re-evaluate our work with the following three major points that we wish to reiterate: (i) The voltage range in the plots are for both the RC filters and the junction. The actual voltage applied on the junction is 50-100 times smaller than the bias voltage applied to the circuit. That is, we are well below the superconducting gap. (ii) In comparison to the work in Ref. 11, we are interested in the voltage range beyond the Josephson regime. However, we are applying voltages much smaller than the superconducting gap which allows to see quartets rather than MARs. We have also presented a scan of S6 showing oscillations of CCC as seen in Ref 11. (iii) One of our major findings is not the flux modulation of resistance itself, but the $0-\pi$ transition and two periodicities ($4e$ and $2e$).

Reviewers' Comments:

Reviewer #1:

Remarks to the Author:

I believe the authors provided an overall reasonable answer to the referee comments, my own included. Since the original review of this manuscript, a new work has appeared on arXiv, that has direct bearing on the issues of quartets visibility in multi-terminal JJs, the role of multiple quantum modes, and multiple Andreev reflection, important points raised by the other two referees: <https://arxiv.org/abs/2201.01373>. That work seems to show a multi-terminal Josephson effect with few quantum modes, and also observes a $V_1=-V_2$ resonance (see Fig. 2 therein), as expected from a quartet process. This new work should be discussed in these contexts. I believe this goes some way toward addressing the points of Referee 3. In particular, the $V_1=-V_2$ line has now been seen in non-graphene junctions with a similar geometry in <https://arxiv.org/abs/2201.01373>.

I also agree with the other two reviewers that the link to Floquet bands and quartets should not be overstated. I think the work is interesting even without overdriving these points beyond what the data supports unequivocally. In particular, resonances at $V_1=-V_2$ can in principle occur due to semiclassical reasons as well (not quartets), as is known in the theory literature on multi-terminal JJs and shown e.g. in Fig. 3 of <https://arxiv.org/abs/2201.01373>.

I recommend the manuscript for publication, provided these additional discussions are incorporated.

Reviewer #2:

Remarks to the Author:

I thank the authors for their response and I do find most of their explanations clear and sufficient for publication. While I still remain skeptical that the suggested Floquet ABS picture is a certainty in their system, the authors have softened the language somewhat, which I believe is appropriate. I further suggest, in the "Emergence of Floquet energy bands" section, to change the wording of the first sentence making it more clear that this is a suggested picture and it is not fully confirmed i.e. "To *better* understand... we ~~need to~~ consider". The language is reasonable in the conclusion. I also ask that they include the number of oscillations/points used in the Fourier analysis in either the figure caption or the main text for transparency.

Reviewer #3:

None

Response to Reviewers' Comments

Reviewer #1:

I believe the authors provided an overall reasonable answer to the referee comments, my own included.

R1.0 We thank Reviewer#1 for the comments in both reviews. We have tried our best to answer all the questions/comments the referee has raised, and we look forward to sharing our results with the community.

1. Since the original review of this manuscript, a new work has appeared on arXiv, that has direct bearing on the issues of quartets visibility in multi-terminal JJs, the role of multiple quantum modes, and multiple Andreev reflection, important points raised by the other two referees: <https://arxiv.org/abs/2201.01373>. That work seems to show a multi-terminal Josephson effect with few quantum modes, and also observes a $V_1=-V_2$ resonance (see Fig. 2 therein), as expected from a quartet process. This new work should be discussed in these contexts. I believe this goes some way toward addressing the points of Referee 3. In particular, the $V_1=-V_2$ line has now been seen in non-graphene junctions with a similar geometry in <https://arxiv.org/abs/2201.01373>.

R.1.1 We thank the reviewer for highlighting this paper to us. This new work (posted during our second round review) indeed is a natural continuation of the evolving field of engineered Andreev bound states (ABS) energy spectrum in order to host tunable topological phases. The authors of this preprint also cite our work (ref. 35 in 2201.01373), saying, "We also observe a signature of Cooper quartet transport (entangled Cooper pairs)[34, 35] indicated by a lower resistance feature along with the line $V_1 = -V_2$." With this new result, now the signature of quartets along the asymmetric bias condition $V_1 = -V_2$ has been seen in multi-terminal JJs with three different materials platforms, including metallic Al-Cu junction, Al-InAs semiconductor wires junction, Al-InAs quantum wells junction, and Al-graphene junctions. We believe that these diverse experiments performed by multiple groups provide the generality of our observation.

In the revision, we have cited the manuscript the reviewer has addressed (now ref. 44) as well as highlighted its relevance to our work: "During the publication period of our work, a complementary transport signature of quartet physics on 2D InAs quantum well heterostructure have been demonstrated - showing the universality of the quartet ABS physics."

2. I also agree with the other two reviewers that the link to Floquet bands and quartets should not be overstated. I think the work is interesting even without overdriving these points beyond what the data supports unequivocally. In particular, resonances at $V_1=-V_2$ can in principle occur due to semiclassical reasons as well (not quartets), as is known in the theory literature on multi-terminal JJs and shown e.g. in Fig. 3 of <https://arxiv.org/abs/2201.01373>.

R.1.2 We thank the reviewer for the comment. As both the first and second referees pointed out that the Floquet picture should be considered as a suggestive interpretation, we have incorporated the change suggested by the second referee. We believe it is now clear the Floquet picture is presented to the reader as a possible interpretation of the results in figure 4.

In the revision, we incorporated the referee's suggestion thereby, clarifying the Floquet picture as a possible model: "To better understand... we suggest considering the following possible scenario...".

3. I recommend the manuscript for publication, provided these additional discussions are incorporated.

R.1.3 We thank the reviewer for the time he/she invested in our manuscript leading to an improvement our work.

Reviewer #2:

I thank the authors for their response and I do find most of their explanations clear and sufficient for publication.

R.2.0 We thank Reviewer#2 for the comments, we are certain it will improve the coherence of our study to the reader.

1. While I still remain skeptical that the suggested Floquet ABS picture is a certainty in their system, the authors have softened the language somewhat, which I believe is appropriate. I further suggest, in the "Emergence of Floquet energy bands" section, to change the wording of the first sentence making it more clear that this is a suggested picture and it is not fully confirmed i.e. "To **better** understand... we \sout{need to} consider". The language is reasonable in the conclusion.

R.2.1 We thank the reviewer for his/hers comment. Following the suggestion of the reviewer, we now soften the Floquet ABS interpretation in our manuscript, stating that this is merely one of the possible suggestions.

In the revision, we incorporated the referee suggestion, thereby clarifying the Floquet picture as a possible model: "To *better understand... we suggest considering the following possible scenario...*".

2. I also ask that they include the number of oscillations/points used in the Fourier analysis in either the figure caption or the main text for transparency.

R.2.2 We thank the reviewer for the comment.

In the revision, we have updated the figure caption to explicitly include the number of oscillations observed in the bare signal used in the FFT calculation: Fig.3 caption in the revision contains: "*a consequence of 8 oscillations in the left panel.*"